

# Collective Monte Carlo updates through tensor network renormalization

**Miguel Frías-Pérez[1,2,3⋆], Michael Mariën[4], David Pérez-García[5,6], Mari Carmen Bañuls[1,2] and Sofyan Iblisdir[3,5]**

**1** Max-Planck-Institut für Quantenoptik, Hans-Kopfermann-Str. 1, D-85748 Garching, Germany
**2** Munich Center for Quantum Science and Technology, Schellingstr. 4, D-80799 München, Germany
**3** Departament de Física Quàntica i Astronomia & Institut de Ciències del Cosmos, Universitat de Barcelona, 08028 Barcelona, Spain
**4** KBC Bank NV - Havenlaan 2 - 1080 Brussels - Belgium
**5** Departamento de Análisis Matemático y Matemática Aplicada, Universidad Complutense de Madrid, 28040 Madrid, Spain
**6** Instituto de Ciencias Matemáticas, Campus de Cantoblanco, 28049 Madrid, Spain

⋆ miguel.frias@mpq.mpg.de

## Abstract

We introduce a Metropolis–Hastings Markov chain for Boltzmann distributions of classical spin systems. It relies on approximate tensor network contractions to propose correlated collective updates at each step of the evolution. We present benchmark computations for a wide variety of instances of the two-dimensional Ising model, including ferromagnetic, antiferromagnetic, (fully) frustrated and Edwards-Anderson spin glass instances, and we show that, with modest computational effort, our Markov chain achieves sizeable acceptance rates, even in the vicinity of critical points. In each of the situations we have considered, the Markov chain compares well with other Monte Carlo schemes such as the Metropolis or Wolff's algorithm: equilibration times appear to be reduced by a factor that varies between 40 and 2000, depending on the model and the observable being monitored. We also present an extension to three spatial dimensions, and demonstrate that it exhibits fast equilibration for finite ferro- and antiferromagnetic instances. Additionally, and although it is originally designed for a square lattice of finite degrees of freedom with open boundary conditions, the proposed scheme can be used as such, or with slight modifications, to study triangular lattices, systems with continuous degrees of freedom, matrix models, a confined gas of hard spheres, or to deal with arbitrary boundary conditions.



# 1   Introduction

Markov Chain Monte Carlo is central to our understanding of strongly correlated systems [1]. When the number of degrees of freedom is too large for exact computations, and perturbative methods are ineffective, Monte Carlo sampling often emerges as the method of choice for numerical investigation. Markov chain Monte Carlo has contributed significantly to the current state-of-the-art in fields like *e.g.* high temperature superconductivity [2], ab initio quantum chemistry [3], or (lattice) quantum chromodynamics [4].

In statistical physics, Monte Carlo sampling has made it possible to chart phase diagrams of several paradigmatic spin systems [5, 6]. The fundamental problem in this context is to sample according to the Boltzmann distribution. To achieve this goal, Markov chain Monte Carlo methods produce a sample by subjecting an initial configuration to a carefully designed stochastic evolution in the space of configurations. Well-known examples are the Metropolis algorithm and heat bath dynamics Markov chains where at most one spin is modified at each step [6], or the Wolff algorithm, where clusters of spins are flipped at once [7, 8]. The applications of these algorithms are countless, but there are important circumstances, such as geometric frustration or disorder, where their limitations become apparent [6, 5].

Over the last two decades, a second notion has been gradually recognised as crucial to our understanding of strongly correlated systems: tensor networks states [9]. In the realm of many-body quantum mechanics, the (simple) entanglement patterns, present in collections of identical particles in short range interaction, enables a description that conceptually transcends mean field approximations, but does not demand the exponential cost of exact diagonalisation [10, 11]. Tensor networks are also used in many-body classical physics. The first applications were proposed by Nishino in [12, 13, 14], and significant developments have been made possible by the advancements in tensor network algorithms. It was shown in [15] that partition functions of all spin systems in nearest neighbour interaction, including inhomogeneous and finite ones, could be represented as a tensor network. While the exact contraction of the tensor network is in general computationally intractable [16, 17], this idea has been used in practice to address many physical problems via an approximate contraction [12, 18, 19, 20, 21, 22, 23, 24, 25]. Tensor network methods have been successfully applied to a variety of classical and quantum two dimensional problems (e.g [20, 26, 27, 28]) including continuous variables [29, 30, 31, 32], and three dimensional classical models [21, 33, 34].

Besides solving concrete problems to very good precision, these contributions have been insightful: we have for example learnt that the notion of bipartition Schmidt weights, ordinary in quantum information theory, is also relevant to classical statistical physics. However, unless an implausible collapse of complexity classes is found, both tensor network and Monte Carlo methods are bound to be ultimately limited, since there exist instances of the Ising model for which the evaluation of the partition function is #P, even in multiplicative approximation [35, 36]. The downside of these fundamental obstructions is that a complete understanding of these systems will (very) likely always be out of reach. The upside is a sustained interest in developing new methods to continually push the boundary of what we can learn about these systems.

Earlier works have looked into particular connections between Monte Carlo and tensor network methods. One perspective has been using Monte Carlo sampling to approximate tensor network contractions, either to contract or optimize a quantum state [37, 38, 39], or to approximate classical partition functions represented by a TN [24]. For the latter task, focusing on the square lattice $O(2)$ model, a thorough comparison between TN-based and Markov Chin Monte Carlo techniques was presented in [40, 41]. A different angle has been the construction of statistical mixtures of pure tensor network states to represent the thermal ensemble of a quantum system [42, 43, 44]. In the context of classical systems, yet another possibility has been proposed in [45, 46], namely the static sampling from a tensor network as a way to obtain relevant spin configurations of a given Hamiltonian at some temperature.

In this work, we present and explore a novel connection between tensor networks and Monte Carlo methods that goes beyond previous studies. Our primary concern here will be *sampling* configurations representative of the Boltzmann distribution of classical nearest neighbour Hamiltonians at finite temperature. To achieve this goal, we introduce a Tensor Network Metropolis-Hastings (TNMH) Markov chain [1, 47], where the asymmetric prior, i.e. the distribution from which the new candidate configuration is drawn at each step, is an approximation to the target distribution, obtained via an inexpensive tensor network renormalisation contraction. Our approach does not oppose but genuinely combines TN and Markov Chain Monte Carlo ideas. In this way, it features concrete advantages with respect to each strategy. The salient properties of the scheme introduced here are the following.

(i) It is *universal*. That is, it works identically for all instances of a given model. This is in contrast to other Monte Carlo algorithms where a powerful prior choice can only be built by relying on a deep insight about the target distribution, and thus has limited applicability beyond the model for which it has specifically been tailored. That is for instance the case of Wolff's algorithm, which performs extremely well for ferromagnetic Ising models, but rather poorly for antiferromagnets or frustrated instances. In turn, we will show that our method fares consistently well for a variety of models that are all very different from one another.

(ii) The scheme produces collective updates. That is, the state of each degree freedom of the considered system is susceptible to change at each Monte Carlo step. We have found that the computational effort scales mildly with increasing acceptance rates in a broad variety of instances. Presumably as a consequence, we have found that the number of Monte Carlo steps necessary to reach convergence is between $\sim 10^1$ and $\sim 10^3$ shorter than those of other well-established Monte Carlo algorithms for several instances of two and three dimensional models of the Ising type.

(iii) As compared to algorithms that purely rely on a tensor network renormalisation of the partition function, the shift to sampling results in the substitution of systematic errors with statistical errors, since our TNMH scheme satisfies the classical sufficient conditions

for convergence (see next section). Thus, modest tensor network contraction schemes, too inaccurate for a direct evaluation of a chosen observable, can be successfully used in our method, as they still enable collective updates with sufficiently high acceptance rates.

(iv) The scheme is versatile. As we shall see, a Markov chain designed for Ising models on a square lattice with open boundary conditions is useful as such to study other systems, such as the $\lambda\phi^4$ model or gases of hard spheres, other interaction graphs such as triangular lattices, and arbitrary boundary conditions.

We have tested our Markov chain systematically in a variety of instances of the Ising model defined on finite square lattices: ferro- and antiferromagnetic, frustrated, disordered, in two and three spatial dimensions. One may anticipate that for systems with large (or even diverging) correlation length, our scheme will perform increasingly poorly if the bond dimension (parameter that controls the cost and accuracy of the tensor network renormalisation) is fixed. Our findings are consistent with this expectation, with drops in acceptance rates actually signaling phase transitions. But we have also observed that for ferro- and antiferromagnetic instances, acceptance rates remain fairly high for systems of considerable size across a phase transition, even with a bond dimension as low as $D = 2$. Equilibration and decorrelation times in our TNMH scheme have been found to be systematically lower than for the Metropolis and Wolff's algorithms. As expected, frustrated and spin-glass instances have turned out to be challenging, not only for fundamental complexity-theoretic reasons, but also because their study is complicated by ill-conditioning issues [48]. However, even in such cases, and without any optimization of our renormalisation procedure, we have observed that acceptance rates stay high enough to be usable down to temperatures that can be considered low by nowadays state-of-the-art standards.

The rest of this paper is organised as follows. The new algorithm is described in section 2 in general terms. In section 3 we explore its performance for two dimensional models. In particular, for a broad variety of instances of Ising models, we explore the role of the bond dimension in the acceptance rates, also in relation to the presence of critical temperatures. We further analyze equilibration and autocorrelation times, and demonstrate how the method can be used to obtain physical observables and chart phase diagrams. In section 4, we demonstrate how the algorithm is also useful for three dimensional systems, and illustrate it for ferromagnetic and antiferromagnetic instances of the Ising model in cubic lattices of up to $16^3$ sites. Section 5 is a discussion of situations where our findings could find further applications, and can be skipped on a first reading; there we discuss triangular lattices, models with continuous variables and systems of hard spheres. An outlook is provided in section 6.

## 2  Markov chains and tensor network renormalisation

In this section, we present a collective Monte Carlo update where, given a current configuration, tensor network renormalisation is used to propose a candidate and to decide whether it should be accepted or not. For the sake of concreteness, and with a view to the example computations that will be considered in the next two sections, we will focus on the Ising model on a square lattice. Generalisation to other nearest neighbour interactions, such as the Potts model, is immediate.

We start by fixing some notation. A lattice will be denoted as $\Lambda = (V, E)$, where $V$ stands for its vertices, and $E$ for its edges. We will focus on systems made of two-state particles (classical spins) residing on the vertices. That is, the sample space will be $\Omega = \{-1, +1\}^{|V|}$. A spin at location $j \in V$ will be denoted by $\sigma_j$.

The Ising Hamiltonian associated with a spin configuration $\omega$ is defined as

$$H(\omega) = -\sum_{i \in V} h_i \sigma_i - \sum_{\langle i,j \rangle \in E} J_{ij} \sigma_i \sigma_j. \tag{1}$$

Our aim is to sample according to the Boltzmann distribution

$$\pi^{(\beta)}(\omega) = e^{-\beta H(\omega)}/Z(\beta), \quad \forall \omega \in \Omega, \tag{2}$$

where $\beta$ denotes the inverse temperature, and

$$Z(\beta) = \sum_{\phi \in \Omega} e^{-\beta H(\phi)}, \tag{3}$$

is the partition function.

A Markov chain is a sequence of configurations $\omega(0), \omega(1), \dots$ where the probability that the $t$-th element of this sequence is in some state $\omega$ only depends on the state of the previous element $\omega(t-1)$, and on some random numbers. That is, a Markov chain is an evolution with short memory. If the process used to determine the state at each time step satisfies some general requirements reminded below, $\lim_{t \to \infty} \omega(t)$ is a state drawn according to the target probability distribution (2). A very simple Markov chain is the celebrated Metropolis algorithm, where at most one spin is flipped at each time step.

A powerful class of Markov chain is that introduced by Hastings in [47]. In the context of Statistical Physics, this class can be described as follows. Given a current spin configuration $\omega$, a candidate configuration $\omega'$ is proposed according to some prior distribution $g^{(\beta)}(\omega'|\omega)$, from which we are able to draw. This candidate is next accepted as the new current state with probability

$$P_{\mathrm{acc}}(\omega \to \omega') = \min\left\{1, \frac{g^{(\beta)}(\omega|\omega')}{g^{(\beta)}(\omega'|\omega)} \times \frac{\pi^{(\beta)}(\omega')}{\pi^{(\beta)}(\omega)}\right\}. \tag{4}$$

As will be shown shortly, this acceptance rule allows to satisfy reversibility (a.k.a. detailed balance), Eq.(8), one of the conditions which when met guarantees convergence to the target probability distribution. If the prior $g^{(\beta)}$ is symmetric in its arguments, the acceptance probability (4) reduces to the celebrated Metropolis algorithm formula. But in some situations, the generalisation proposed by Hastings allows encoding some information about the target distribution in the possibly asymmetric prior $g^{(\beta)}(\omega'|\omega)$, in a beneficial way. It can for example result in a boosted exploration of the sample space along the iterations of the Markov chain. The Swendsen-Wang and the Wolff cluster algorithms are examples of Metropolis-Hastings construction [7, 8]. Actually, an ideal prior is one where $g^{(\beta)}(\omega'|\omega) = \pi^{(\beta)}(\omega')$, that is, the prior that consists in direct sampling according to the target probability distribution $\pi^{(\beta)}$. Of course, for generic instances of the Ising model, such an ideal prior is unavailable. But an approximation $\tilde{\pi}^{(\beta)}$ to this ideal prior might be good enough for Monte Carlo. We will be concerned with such approximations that can be constructed through tensor network renormalization.

Let $n = |V|$ represent the system size, and let $\{1, 2, \dots, n\}$ denote a certain sequential labelling of the vertices (see e.g. figure 19 in App. A). Using Bayes formula, the Boltzmann distribution can be expressed as

$$\pi^{(\beta)}(\omega) = \pi_1^{(\beta)}(\sigma_1) \prod_{k=2}^{n} \pi_k^{(\beta)}(\sigma_k | \sigma_1 \dots \sigma_{k-1}), \tag{5}$$

where $\pi_1^{(\beta)}$ stands for the marginal distribution of the first spin, and $\pi_k^{(\beta)}(\cdot|\sigma_1 \dots \sigma_{k-1})$ denotes the conditional distribution for the $k$th spin when the spins 1 through $k-1$ are fixed to values

$\sigma_1, \ldots \sigma_{k-1}$. The marginal distribution for the first spin $\pi_1^{(\beta)}(\sigma_1)$ can be expressed as the ratio of two partition functions: $\pi_1^{(\beta)}(\sigma_1) = Z(\beta|\sigma_1)/Z(\beta)$, where $Z(\beta|\sigma_1)$ represents the partition function for a system with the same nearest neighbour Hamiltonian as in $Z(\beta)$ but where the first spin has been fixed to the value $\sigma_1$. As mentioned in the introduction, the partition function (3) of any nearest neighbour Hamiltonian can be expressed exactly as a tensor network (TN), whose bond dimension is equal to the number of states accessible by each local degree of freedom. For the Ising model, this number is equal to two. In general, neither $Z(\beta|\sigma_1)$ nor $Z(\beta)$ can be evaluated exactly. But a TN renormalisation scheme yields approximations $\widetilde{Z}(\beta|\sigma_1)$, and $\widetilde{Z}(\beta)$ for each of these quantities (see Appendix A). With them, one can construct an approximation $\widetilde{\pi}_1^{(\beta)}(\sigma_1) = \widetilde{Z}(\beta|\sigma_1)/\widetilde{Z}(\beta)$ to the true marginal distribution for the first spin. This Bernoulli distribution is next sampled. Let $s_1$ the outcome obtained. With this fixed value for the first spin, one can compute an approximation $\widetilde{Z}(\beta|s_1, \sigma_2)$ of the partition function for each value $\sigma_2$ for the second spin. These approximations are then used to construct an approximation $\widetilde{\pi}_2^{(\beta)}(\sigma_2|s_1) = \widetilde{Z}(\beta|s_1, \sigma_2)/\widetilde{Z}(\beta|s_1)$ to the distribution for the second spin, conditioned on the value $s_1$ for the first spin. This second Bernoulli distribution is then sampled. And so on. For all other sites $k > 2$, the conditional probability distribution $\pi_k^{(\beta)}(\sigma_k|s_1 \ldots s_{k-1})$ can be expressed as the ratio of two TN contractions $Z(\beta|s_1 \ldots s_{k-1}\sigma_k)/Z(\beta|s_1 \ldots s_{k-1})$, and a TN renormalisation scheme provides approximations $\widetilde{Z}(\beta|s_1 \ldots s_{k-1})$ and $\widetilde{Z}(\beta|s_1 \ldots s_{k-1}\sigma_k)$ to $Z(\beta|s_1 \ldots s_{k-1})$ and $Z(\beta|s_1 \ldots s_{k-1}\sigma_k)$ respectively. These approximations are in turn used to compute an approximation $\widetilde{\pi}_k^{(\beta)}(\sigma_k|s_1 \ldots s_{k-1})$ to $\pi_k^{(\beta)}(\sigma_k|s_1 \ldots s_{k-1})$, which is sampled and yields an outcome $s_k$. Fig. 1 illustrates the first two steps of this sequential sampling. The configuration $(s_1, \ldots, s_n)$ obtained after the whole lattice is swept will have been drawn with probability

$$\widetilde{\pi}^{(\beta)}(s_1, \ldots, s_n) \equiv \widetilde{\pi}_1^{(\beta)}(s_1) \prod_{k=2}^{n} \widetilde{\pi}_k^{(\beta)}(s_k|s_1 \ldots s_{k-1}), \tag{6}$$

which the identity (5) shows to be an approximation to $\pi^{(\beta)}(s_1, \ldots, s_n)$. We will be interested in schemes where the Metropolis-Hastings probability to select a candidate $\omega'$ reads:

$$g^{(\beta)}(\omega'|\omega) \equiv \widetilde{\pi}^{(\beta)}(\omega'). \tag{7}$$

As explained in Appendix A, the approximate probability $\widetilde{\pi}^{(\beta)}(\omega)$ can be evaluated for any configuration $\omega$, and the update rule (4) can be implemented. Our construction is summarized in Algorithm 1.[1]

## Properties of the TNMH Markov chain

(i) The construction is *universal* in the sense that it is independent of the magnetic fields and couplings that define the Ising instance being considered. Yet, it is *adaptive* in that the details of the Hamiltonian are taken into account when the tensors are constructed.

(ii) The constitution of the candidate is independent of the current configuration.

(iii) The update (7) is *collective and correlated*: in principle *all* spins of the system could be refreshed in a single Monte Carlo step, and the spin values proposed at different sites are conditioned by the correlations present in the tensor network. We believe this feature is the principal cause for the high acceptance rates and fast equilibration reported in the next section. Whereas a local update rule could have a hard time overcoming energy

---

[1]After completion of our work, we were made aware that a similar sampling scheme, based on Bayes' chain rule, has been proposed to directly sample an approximation to the Gibbs distribution [45]. But no study on how to use it in a Metropolis-Hastings Markov chain was made in that work.

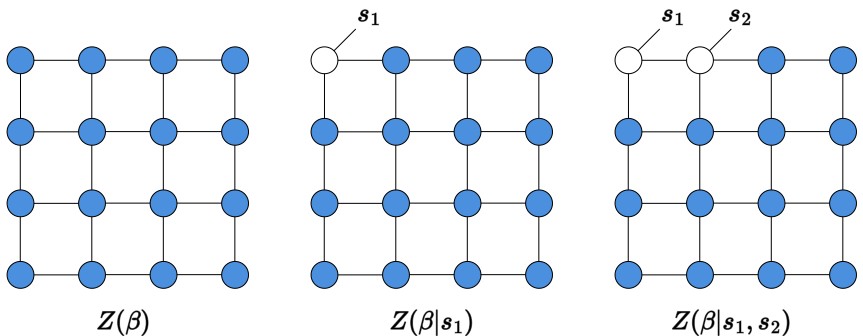

Figure 1: Pictorial illustration of the first two steps of the TNMH sequential sampling. White dots refer to sites where the spin value has been fixed.

barriers, we expect our algorithm to be more capable of hopping between distant regions of the configuration space and escape local minima in a single iteration of the Markov chain.

(iv) The transition matrix, i.e. the set of probabilities to transition from a configuration $\omega$ to a configuration $\omega'$, $\mathcal{T}(\omega \to \omega') = \widetilde{\pi}^{(\beta)}(\omega') \times P_{\text{acc}}(\omega \to \omega')$, is reversible (i.e. it satisfies detailed balanced):

$$\pi^{(\beta)}(\omega)\,\mathcal{T}(\omega \to \omega') = \pi^{(\beta)}(\omega')\,\mathcal{T}(\omega' \to \omega). \tag{8}$$

That is, the expression

$$\pi^{(\beta)}(\omega)\widetilde{\pi}^{(\beta)}(\omega')\min\left\{1, \frac{\widetilde{\pi}^{(\beta)}(\omega)}{\widetilde{\pi}^{(\beta)}(\omega')} \times \frac{\pi^{(\beta)}(\omega')}{\pi^{(\beta)}(\omega)}\right\},$$

is manifestly symmetric in $\omega$ and $\omega'$.

Furthermore, when numerical errors are small enough that all the conditioned partition functions $\widetilde{Z}(\beta|\sigma_1 \ldots \sigma_k)$ are strictly positive (see Appendix A), the Markov chain is also irreducible:

$$\mathcal{T}(\omega \to \omega') > 0, \quad \forall \omega, \omega' \in \Omega.$$

Thus, even if the distributions $\{\pi_k^{(\beta)} : k \in V\}$ turned out to be poorly approximated by the TN renormalisation scheme used, it is still possible to guarantee that the Markov chain will eventually converge to the target probability distribution [49]. This last point will be illustrated with three-dimensional Ising models.

---

**Algorithm 1** TNMH Markov chain

---

1: Compute the tensors associated with the distribution (2).
2: Set $t = 0$, and draw some initial configuration $\omega(0)$ according to any distribution over $\Omega$.
3: If $t > t_{\max}$ go to 8.
4: Use the tensor network to draw a candidate configuration $\omega'$ according to Eq.(7).
5: Evaluate the probabilities $\widetilde{\pi}^{(\beta)}(\omega(t))$ and $\widetilde{\pi}^{(\beta)}(\omega')$.
6: Accept the change $\omega(t) \leftarrow \omega'$ with probability $\min\left\{1, \frac{\widetilde{\pi}^{(\beta)}(\omega|\omega')}{\widetilde{\pi}^{(\beta)}(\omega'|\omega)} \times \frac{\pi^{(\beta)}(\omega')}{\pi^{(\beta)}(\omega)}\right\}$.
7: $t \leftarrow t + 1$. Go to 3.
8: End.

---

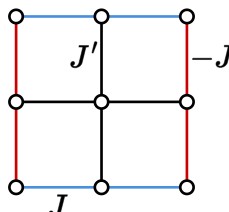

Figure 2: Distribution of the couplings on the $J'-J$ model. Black lines indicate bonds with a coupling of $J'$, while red and blue bonds have couplings $-J$ and $J$, respectively.

Before turning to applications of Algorithm 1, there are two points we would like to stress. (i) Although the construction of a candidate at each iteration of the Markov chain is actually independent from its current configuration, the decision to accept this candidate *does* depend on the current state. That is, the transition probability $\mathcal{T}(\omega \to \omega')$ is *not* independent of $\omega$. This can be seen explicitly:

$$\mathcal{T}(\omega \to \omega') = \widetilde{\pi}^{(\beta)}(\omega') \min\{1, \frac{\widetilde{\pi}^{(\beta)}(\omega)}{\widetilde{\pi}^{(\beta)}(\omega')} \times \frac{\pi^{(\beta)}(\omega')}{\pi^{(\beta)}(\omega)}\}.$$

Notice that a similar sampling scheme, based on Bayes' chain rule, has been proposed to directly sample an approximation to the Gibbs distribution [45]. In constrast, TNMH is not an approximate Gibbs sampler: the decision to accept or reject the candidate, depending on the Metropolis-Hastings ratio, marks an essential difference. (ii) Even though the tensor network contractions used in TNMH are (inevitably) approximate, the reversibility condition is *exactly* satisfied. This ensures the asymptotic convergence to the Gibbs distribution.[2]

## 3 Two-dimensional Ising models

In order to assess the potential of the construction presented in the previous section, we have run tests on instances of the two-dimensional Ising models chosen to cover a broad range of cases ($L \times L$ square lattice):

- Ferromagnetic: $J_{ij} = 1 \; \forall \; \langle i,j \rangle$, $h_i = 0 \; \forall i$.

- Antiferromagnetic: $J_{ij} = -1 \; \forall \; \langle i,j \rangle$, $h_i$ constant across the whole lattice.

- $J'-J$ model: In this model $h_i = 0 \; \forall i$, and couplings alternate between even and odd rows or columns (see Fig.2):

$$J_{2j-1,k} = J', \quad J_{2j,k} = J,$$
$$J_{k,2j-1} = J', \quad J_{k,2j} = -J, \quad j = 1, \dots, L/2.$$

The point $J = J'$, known as the fully frustrated square lattice Ising model (or Villain model), is characterised by extensive ground state degeneracy and maximal frustration.

- Edwards-Anderson spin glass: this disordered model is such that $h_i = 0 \; \forall i$, and $J_{ij}$ are random couplings sampled from a Gaussian distribution with zero mean and unit variance [50].

---

[2] We are grateful to our anonymous referees for discussions that convinced us these two points should be emphasised.

We will be interested in the following observables: the energy per spin,

$$\varepsilon = \frac{1}{|V|Z(\beta)} \sum_{\omega \in \Omega} H(\omega)\, e^{-\beta H(\omega)},$$

the magnetisation density,

$$m = \frac{1}{|V|Z(\beta)} \sum_{\omega \in \Omega} \left| \sum_{i \in V} \sigma_i \right| e^{-\beta H(\omega)},$$

the staggered magnetisation density,[3]

$$m_s = \frac{1}{|V|Z(\beta)} \sum_{\omega \in \Omega} \left| \sum_{i \in V} \text{sign}(i)\, \sigma_i \right| e^{-\beta H(\omega)},$$

where $\text{sign}(i)$ is equal to $\pm 1$ in a checkerboard manner. Finally, we will consider also the magnetic susceptibility, defined for a system with a uniform magnetic field as

$$\chi = \partial m / \partial h.$$

Unless stated otherwise, we will be considering open boundary conditions.

**Role of the bond dimension**

A crucial ingredient of the algorithm described in the previous section is the substitution of exact partition functions $Z(\beta|\sigma_1 \dots \sigma_{k-1})$ with approximations $\widetilde{Z}(\beta|\sigma_1 \dots \sigma_{k-1})$ obtained by tensor network renormalisation. Amongst all availables methods for this renormalisation, we have used the matrix product state (MPS) renormalisation scheme described in [15] (see also Appendix A). It is a choice of simplicity, which turned out to be sufficient for our purposes. We however would like to stress that the analogous TNMH Markov chain can be defined using any other contraction scheme, and some might yield better results than those presented here. In MPS renormalisation, both the accuracy of the approximation and the computational effort increase with an integer parameter, the bond dimension, commonly denoted $D$. We thus expect the total variation distance between the target and the prior distribution,

$$\left\| \pi^{(\beta)} - \widetilde{\pi}^{(\beta)} \right\|_{\text{TV}} = \frac{1}{2} \sum_{\omega \in \Omega} \left| \pi^{(\beta)}(\omega) - \widetilde{\pi}^{(\beta)}(\omega) \right|, \tag{9}$$

to decrease with increasing values of $D$. As a result, Monte Carlo rejection rates should decrease as the bond dimension grows large.

To characterize the behaviour of our method, we have explored the interplay between the bond dimension, the temperature and the rejection rate for the four different models mentioned above (Fig. 3). In all cases, we have verified that the rejection rate decreases with increasing bond dimension, and even modest values of the bond dimension may yield virtually rejection-free updates. At the same time, for a fixed $D$, rejection rates increase in the vicinity of critical points. This can be understood considering that, presumably, the distance $\|\pi^{(\beta)} - \widetilde{\pi}^{(\beta)}\|_{\text{TV}}$ for fixed $D$ will increase with the true correlation length.

Fig. 3a demonstrates these features for the ferromagnetic case. This model exhibits, in the limit of large system sizes, a second-order phase transition at $T_c = 2/\log(1+\sqrt{2}) \approx 2.269$ from

---

[3]When the external magnetic field is uniformly naught, averaging over Monte Carlo samples would result in zero (staggered) magnetisation even at temperatures where the system is known to exhibit a finite spontaneous magnetization. The absolute values appearing in our definition of the (staggered) magnetisation are meant to counter this artefact.

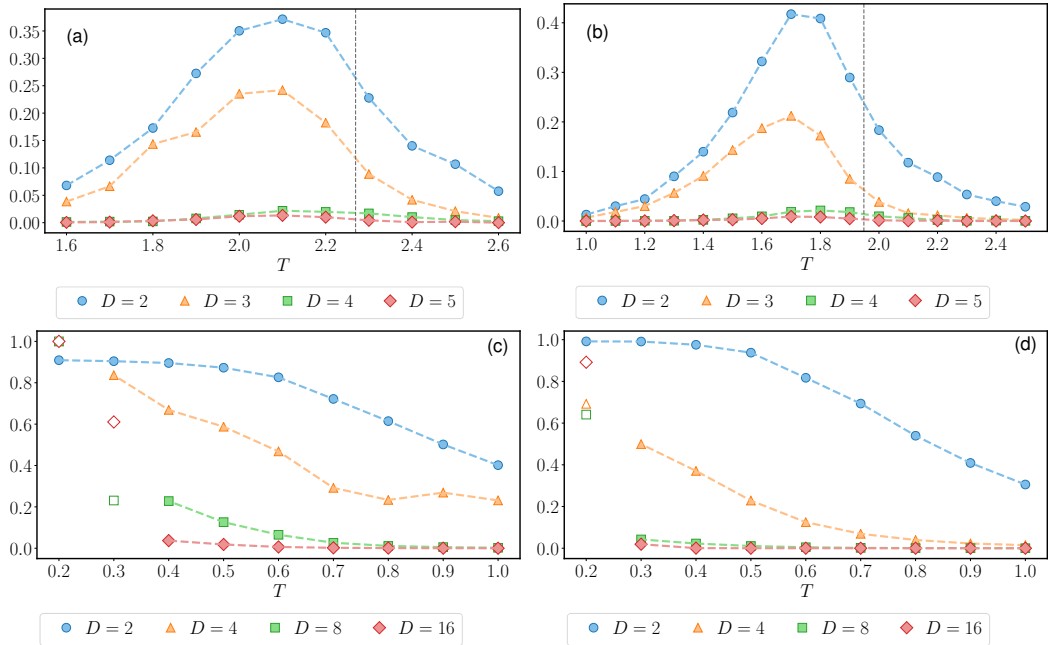

Figure 3: TNMH rejection rates as a function of the temperature for four different instances of the Ising model at a fixed system size: (a) Ferromagnetic, (b) Antiferromagnetic with a constant magnetic field $h = 2$, (c) Fully frustrated, (d) Gaussian spin glass. The geometry is that of a $32 \times 32$ square lattice with open boundary conditions in all four cases. For each model, the rejection rate was obtained by averaging over 40 independent chains, each run for a model-dependent number of steps. For the fully frustrated and spin glass cases, the points at temperatures where we believe our method starts to suffer from ill-conditioning issues are indicated by empty markers. For the ferro- and antiferromagnetic instances, the vertical lines indicate criticality in the thermodynamic limit.

a magnetically ordered to a paramagnetic phase [52]. Still, for the system size considered, $32 \times 32$, rejection rates remain remarkably low (below 0.4) across the whole temperature range that surrounds the critical temperature. Actually, even a bond dimension as low as $D = 2$ appears to already be sufficient to achieve our goal of producing collective updates frequently. More details regarding the vicinity of the critical point are provided on Fig. 4a, where we have plotted the rejection rate as a function of the system size for different bond dimensions. Even for systems as large as $256 \times 256$, acceptance rates of about 0.4 can be obtained using only a bond dimension $D = 4$. As can be appreciated from the inset of this figure, our data suggest that the bond dimension only needs to grow *logarithmically* with the system size in order to maintain the acceptance rate above a threshold value. Our observations for the antiferromagnetic case (Fig. 3b) are similar.

For the fully frustrated case of the $J' - J$ model (Fig. 3c) we obtain lower acceptance rates, as compared to the two previous cases, but still high enough that the Markov chain is usable down to $T = O(10^{-1})$. As expected, the rejection rate increases when approaching the $T = 0$ critical point. Still, the minimal cost curve $D = 2$ is sufficient to obtain decent acceptance rates down to at least $T = 0.2$, and increasing the bond dimension again suppresses rejection events. At very low temperatures, acceptance rates drop dramatically and numerical instabilities typical of frustrated systems pointed out in [48] set in. Some strategies exist to mitigate these effects, but their discussion is beyond the scope of the present work, and will be

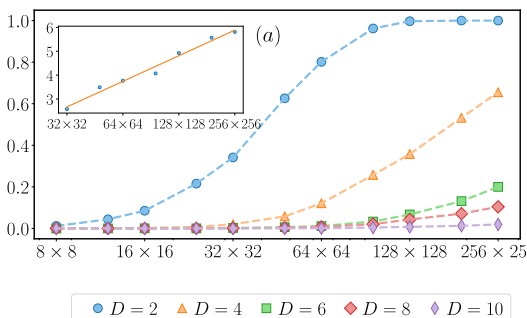
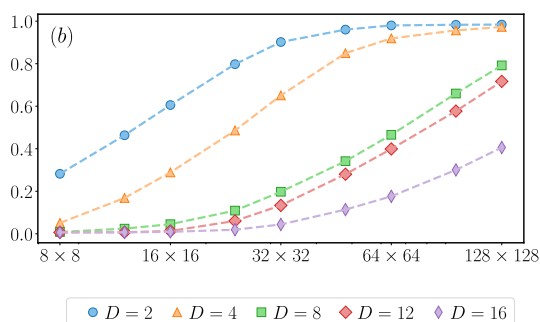

Figure 4: (a) TNMH rejection rates near the ferromagnetic Ising phase transition ($T \simeq 2.27$) as a function of the system size for different bond dimensions. Dashed lines are simply a guide to the eye. Inset: Bond dimension needed to maintain a fixed rejection rate (in this case 0.25, although the behaviour seems to be independent of the value chosen) as a function of the system size. The fit shows that the increase in the bond dimension seems to be only logarithmic. (b) TNMH rejection rates for the fully frustrated Ising model at $T = 0.4$ as a function of the system size for different bond dimensions.

the subject of a separate study [53]. Again, we have looked at the rejection rate as a function of the system size for different bond dimensions (see Fig. 4b). Even though this instance is more challenging, the example in the figure demonstrates that at $T = 0.4$ perfectly usable acceptance rates of about 0.2 or higher can be obtained for systems of size up to $128 \times 128$ using a bond dimension, $D = 6$, for which computations are not too demanding. Fig. 5 provides additional data regarding the $J'-J$ model, beyond the fully frustrated point $J = J'$. It is remarkable that the observed maxima of rejection rates are consistent with the predicted critical lines of this model.

Our findings for the Edwards-Anderson spin glass, Fig. 3d, are qualitatively similar to those for the fully frustrated case, presumably because this spin glass is also critical at $T = 0$ [50].

Improved approximations of the contraction will generally result in a higher acceptance rate. But actually, as far as this acceptance rate does not vanish and scales well with the system size, the TNMH scheme should be applicable.

**Equilibration and decorrelation**

Equilibration and auto-correlation times are the two crucial time scales in Monte Carlo simulations. The former controls the number of steps needed by the Markov chain to decouple from the initial distribution (that is, the distribution from which the first configuration of the chain is sampled) and reach the desired equilibrium distribution. The latter determines the minimal time between two consecutive sample extractions in order to guarantee statistical independence. Since these times are typically difficult to bound, let alone calculate, rigorously, heuristic diagonostics are commonly used to estimate them. We have used two such heuristics to provide evidence that these two time scales are relatively short for the TNMH scheme of Section 2.

A standard technique to decide that equilibration has occurred is to monitor an observable from its value at the beginning of the Markov chain until it appears to plateau at an equilibrium value around which it fluctuates [55]. Fig. 6a illustrates the evolution of the expectation value for the magnetisation of a ferromagnet, and independent Markov chains evolved according to either the TNMH algorithm 1 (blue), a simple spin flip Metropolis algorithm (green) or Wolff's

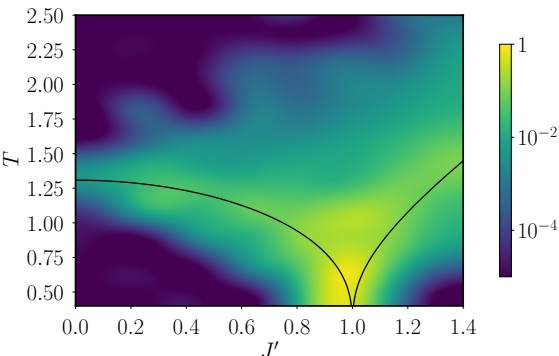

Figure 5: TNMH rejection rate for the $J' - J$ model as a function of the temperature and the value of one of the couplings (the other has been set to unity). Computations made with a bond dimension $D = 4$. Rejection rates obtained as averages over 40 independent Markov chains, each run for 200 steps. Lattice dimensions: $32 \times 32$. The phase separatrix predicted in [54] is shown in black.

cluster algorithm (orange) – which is known to perform best for ferromagnetic instances. We see in this numerical experiment that the number of time steps required for TNMH to equilibrate is about $1/80$ to $1/40$ the number of steps required for Wolff algorithm, and about $1/10^3$ the number of sweeps required by the single spin flip Metropolis algorithm.

More sophisticated equilibration diagnostics can be devised for specific problems. In particular, for the Edwards-Anderson spin glass, we have run the following test, discussed in [56, 57]. Let $\langle X \rangle$ stand for the thermal average of an observable $X$, and $[x]_{\text{av}}$ denote the disorder average of a quantity $x$ that might depend on the coupling constants $\{J_{ij}\}$. We have considered the disorder averaged energy.

$$[\langle H \rangle]_{\text{av}} = -\int \prod_{\langle ab \rangle} \frac{dJ_{ab}}{\sqrt{2\pi}} e^{-J_{ab}^2/2} \sum_{\langle ij \rangle} J_{ij} \langle \sigma_i \sigma_j \rangle . \tag{10}$$

At equilibrium, integrating by parts allows to prove that

$$\Delta \equiv \frac{1}{|V|} [\langle H \rangle]_{\text{av}} + \frac{1}{|V|} \beta \left( |E| - \sum_{\langle ij \rangle} [\langle \sigma_i \sigma_j \rangle^2]_{\text{av}} \right) = 0 , \tag{11}$$

where $\sum_{\langle ij \rangle} [\langle \sigma_i \sigma_j \rangle^2]_{\text{av}}$ is a quantity known as the link overlap. Starting configurations of Markov chains, drawn according to an easy distribution, typically have high energy and small link overlap. As a result, $\Delta$ typically has a non-zero value when the Markov chain is started. As Monte Carlo steps are taken, this value decreases in magnitude. It is a common heuristic to decide equilibration has occurred when $\Delta$ is below a given threshold value.

We have implemented this heuristic test of convergence with $10^4$ disorder realisations, using the TNMH Markov chain. We have observed that after a few time steps, $\Delta$ drops drastically from its initial value before stagnating at a small but finite value. For the vast majority of disorder realisations, the Markov chain behaved well. That is, it shows frequent jumps between different configurations from one time step to the next. However, for some disorder realizations, a few Markov chains (typically less than a sixth of the chains we simulate for a given disorder realization) remained stuck in their original configurations. This inertia is what causes $\Delta$ to stagnate. Increasing the bond dimension did not help. Actually, we believe the issue is rather related to an ill-conditioning of the tensor network contraction, induced by frustration, an effect previously reported in Ref. [48]. As a result, the approximate probability

Table 1: First row: target value of $\Delta$, as defined by Eq.(11). Next rows: each entry represents the number of lattice sweeps necessary to decrease $\Delta$ below the value indicated in the same column for the given algorithm. An entry of the form '$> a$' indicates that more than $a$ iterations are needed to reach the desired value of $\Delta$. The system considered is the same as in Fig. 6.

| $\Delta$ | 0.25 | 0.15 | 0.05 | 0.025 |
|---|---|---|---|---|
| Metropolis | $> 10^5$ | $> 10^5$ | $> 10^5$ | $> 10^5$ |
| PT | $> 10^5$ | $> 10^5$ | $> 10^5$ | $> 10^5$ |
| PT + ICM | $1.1 \cdot 10^3$ | $1.6 \cdot 10^3$ | $3.2 \cdot 10^3$ | $4.3 \cdot 10^3$ |
| TNMH | $> 10^2$ | $> 10^2$ | $> 10^2$ | $> 10^2$ |
| TNMH + Metropolis | 3 | 3 | 5 | 5 |

weights can be off their true value by orders of magnitude.[4] As can be seen from Eq.(4), this mismatch affects the acceptance rates. That is, when the Markov chain hits a configuration $\omega$ such that $\widetilde{\pi}^{(\beta)}(\omega)/\pi^{(\beta)}(\omega) \ll 1$, it may remain stuck for a long time, as we have observed.

Various strategies are possible to mitigate the effect of ill-conditioning. One is to work with greater machine precision, using for example the techniques described in Ref. [58]. Another is to consider a variation of the TNMH Algorithm 1. A very simple such variation consists in interspersing spin flip Metropolis sweeps in between TNMH moves. We have tested this possibility. Our findings are reported on Fig. 6b and on Table 1, where we compare our times with state-of-the-art methods: parallel tempering (PT) and parallel tempering combined with isoenergetic cluster moves (PT + ICM) [56, 57]. This comparison clearly shows that the combination of TNMH with single flip MC sweeps allows us to outperform these methods by orders of magnitude. This result is interesting: whereas both TNMH and the Metropolis algorithm show poor performance individually (for different reasons; ill-conditioning in the case of TNMH, locality in the case of single flip updates), their alternating use is drastically more efficient than either of them.

So far, we have counted equilibration times in steps of the Markov chain, which has conceptual relevance. From a practical point of view though, it is also interesting to know how the TNMH compares to other methods when looking at program execution times. To make such a comparison fairly is a delicate issue because we have not sought to optimise our code at all: a detailed comparison with e.g. Metropolis sweeps, which is simpler to optimise is a project in itself. We can however provide *indicative* times related to the data presented on Table 1. With our setup, the times to get to $\Delta < 0.025$ are $> 3.93 \times 10^7$ sec, $1.76 \times 10^6$ sec, and $6.75 \times 10^4$ sec for the parallel tempering method, the parallel tempering method supplemented with isoenergetic cluster moves, and TNMH respectively. We believe that these estimates credibly signal the *practical* potential of the TNMH Markov chain introduced here.

We next move to autocorrelation times, that is, after equilibration is reached, the time needed between two sample extractions to guarantee (sufficient) independence. Given an observable $X$, we study the time correlation function

$$C_X(t) = \frac{\langle X(t_0)X(t_0 + t)\rangle - \langle X(t_0)\rangle\langle X(t_0 + t)\rangle}{\langle X^2(t_0)\rangle - \langle X(t_0)\rangle^2}, \tag{12}$$

where $t_0$ is assumed larger than the equilibration time. As discussed in [55], at large $t$, we expect $C_X(t)$ to decay exponentially, with a time scale set by the decorrelation time. As the

---

[4]Interestingly, it is not clear that such configurations actually correspond to local minima or maxima of the energy.

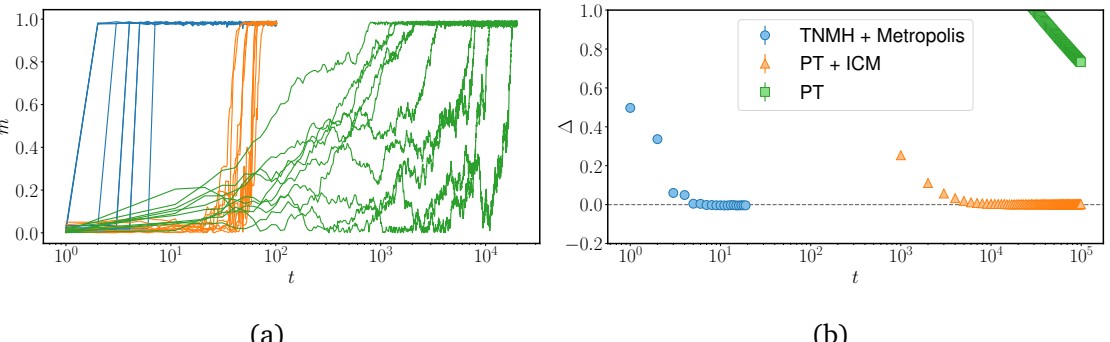

(a)                                    (b)

Figure 6: (a) Absolute value of the magnetization per site along different chains at $T = 1.5$, for three algorithms, $D = 4$ TNMH (blue), Wolff's algorithm (orange) and a simple spin flip (green) for a ferromagnetic $64 \times 64$ lattice. Each line represents an independent run. Time $t$ is measured in Markov chain iterations for TNMH, in sweeps for the simple spin flip Metropolis, and in cluster updates for the Wolff algorithm. (b) Difference between the (disordered averaged) energy per spin computed from the Hamiltonian and computed from the link overlap, Eq. (11), as a function of the number of iterations for three algorithms: TNMH + Metropolis (blue), (PT + ICM) (orange) and PT (green). Bond dimension for the TNMH moves: $D = 16$. Lattice dimensions: $32 \times 32$ (open boundary conditions). The symbol $t$ represents the number of iterations for TNMH, and the number of lattice sweeps for PT and PT+ICM. The error bars, smaller than the symbols, have been computed by estimating the variance of the disorder. The number of Markov chains used for the thermal average is 30 in all cases, the number of different instances used for the disordered average is $10^4$ and the temperature has been set to $T = 0.212$. (b)

exponential tail has a fixed amount of noise, controlled by the number of samples, a useful measure to determine that time scale is the integrated correlation time

$$\tau_X^{\text{int}}(t) \equiv 1 + 2 \sum_{t'=1}^{t} C_X(t'). \tag{13}$$

It can also be shown that it is approximately the factor that enhances the variance when averaging over samples that are not sufficiently decorrelated [55]. The two quantities are plotted on Fig. 7a for the magnetization of a Fully Frustrated Ising model on a $32 \times 32$ lattice at $T = 1$. The motivation for choosing this observable is that often the energy is a poor choice to measure the decorrelation of samples in a Markov chain. At low temperatures, it is impossible to distinguish global changes in a configuration of a Markov chain from local motion around a local minima just by tracking the energy. In this particular case, choosing an observable that breaks the global spin flip symmetry of the model in consideration allows to assess the ergodicity of the scheme, since for any configuration with energy $E$ and magnetization $m$ there exists another with magnetization $-m$ and same energy. The data shown in Fig. 7a shows that TNMH outperforms a local algorithm by almost two orders of magnitude. Furthermore, we expect that the difference in performance can only increase as the temperatures are lowered or the system size is increased.

To further illustrate sample-to-sample decorrelation in our algorithm, we have considered the fully frustrated Ising model and represented in Fig. 7b snapshots at different times for TNMH and for Metropolis sweeps starting from a same configuration. The difference between both techniques is striking: while the configurations appearing in our technique seem to bear no resemblance to one another from one acceptance to the next, memory of the initial config-

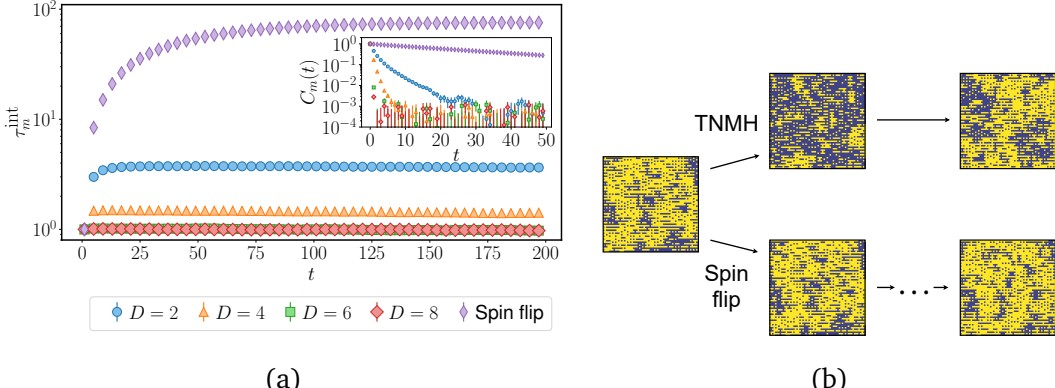

(a)  (b)

Figure 7: (a) Integrated correlation time (13) and (inset) decay of the autocorrelation function (12) of the magnetization as a function of time, for different bond dimensions on a Fully Frustrated Ising model on a $32 \times 32$ lattice with open boundary conditions at $T = 1$. The error bars have been computed by estimating the variance of the observables using the same samples. (b) Snapshots of the evolution of two different Markov chains starting from the same configuration, for the fully frustrated Ising model on a $64 \times 64$ lattice at $T = 0.5$. The top plots display the configuration obtained after three consecutive steps of the TNMH method (with bond dimension $D = 8$), while those below show configurations after Metropolis sweeps at times $t = 0, 1, 10$.

uration can still be appreciated visually after ten Metropolis sweeps.

## Observables

We now turn to the estimation of observables from the samples output by the TNMH Markov chain. We have focused on the ferromagnetic case and the antiferromagnetic one with an external field. The absolute value of the magnetisation for the ferromagnetic case is plotted on Fig. 8a (inset) and is in good agreement with the theory [52]. Fig. 8a also shows estimates for the fourth order Binder cumulant [6], $g = (3 - \langle m^4 \rangle / \langle m^2 \rangle^2)/2$. One can appreciate that the phase transition point is correctly signalled by the locus where all data sets meet, as expected. On Fig. 8b, we have represented the staggered susceptibility as a function of the temperature and the external magnetic field for an antiferromagnet. Our findings seem to be in good agreement with previous studies of this model [59, 60]. When the external field is naught, the ferromagnetic phase transition around $T_c = 2/\ln(1 + \sqrt{2})$ is recovered, as expected, since for a square lattice, a local change of variables allows a mapping between antiferro and ferromagnetic instances of the Ising model. As the field increases, the temperature at which the phase transition takes place decreases. The intuition for this fact is as follows: at $h = 0$ and below the critical temperature one has antiferromagnetic order. In the large $h$ limit at the same temperature all spins would align with the external field and one would have ferromagnetic order. Thus, some phase boundary must be encountered when going from one to the other.

## 4  Three-dimensional Ising models

Just as for planar systems, the partition function of a three-dimensional Ising model can be expressed as a tensor network. As a consequence, our TNMH algorithm immediately extends to three dimensions. The approximate contraction of a three-dimensional TN is however a more

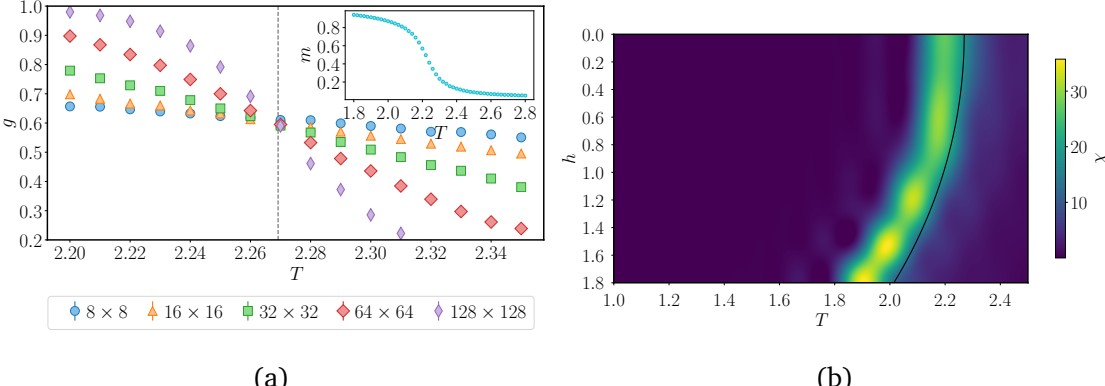

(a)  (b)

Figure 8: (a) Binder ratio as a function of temperature for the two dimensional ferromagnetic Ising model obtained through TNMH with a bond dimension $D = 6$. The data approximately cross at one point, signalling a phase transition, in great concordance with the theoretical result $T \approx 2.269$. Inset: Magnetisation of the ferromagnetic Ising model on a square $64 \times 64$ lattice with open boundary conditions. The error bars have been computed via a jackknife analysis for the Binder cumulant and by estimating the variance for the magnetization, and are smaller than the symbols. (b) Susceptibility $\chi$ for an antiferromagnetic Ising model with an external field, on a $64 \times 64$ square lattice obtained through TNMH with a bond dimension $D = 6$. Black: theoretical prediction of the critical line in the thermodynamical limit.

demanding problem than its two-dimensional analogue. Still, TN renormalisation schemes can be applied to find an approximation to the contraction [21, 33, 34]. We have chosen an unsophisticated renormalisation scheme involving projected entangled pair states (PEPS). Two cutoff parameters now govern the effort put in the TNMH for this implementation: a boundary PEPS bond dimension $D$, and a boundary MPS bond dimension $\chi$ (see Appendix A for details). We have considered two instances of the Ising model: ferromagnetic, and antiferromagnetic with an external magnetic field. The upshot is that our TNMH performs very well, even with rather low values for $D$ and $\chi$.

Fig. 9 shows the interplay between the two parameters $D$ and $\chi$, the temperature, and the rejection rate. Again, the peak in the rejection rate signals the presence of a critical point (displaced due to finite size effects). As this critical point is approached, rejection rates increase much faster than in two dimensions, and putting in more computational effort by increasing $D$ and $\chi$ now produces milder drops in rejection rates. We attribute this situation to an increase of correlations in the system due to a higher coordination number for each spin. Still, these preliminary results are very encouraging, since using a non-optimized contraction scheme, and modest values for the parameters $D$ and $\chi$, usable acceptance rates ($> 0.12$ and $> 0.05$ for the ferro- and antiferromagnetic case respectively) have been found across the whole temperature range considered, for systems as large as $16^3 = 4096$ spins.

Analogous to Fig. 6a, which explored equilibration in the two dimensional case, we show the energy of a $16^3$ ferromagnetic Ising model as a function of time on Fig. 10, both for the TNMH Markov chain and for the three-dimensional Wolff algorithm. As in two dimensions, the former appears to necessitate a lower number of steps than the latter. The magnetisation of the ferromagnetic Ising model has also been plotted in Fig. 11 and shows good agreement with previous studies [61, 62, 63].

Since observables can also be expressed as a TN, it is possible to estimate them using a direct contraction [15], and one might then wonder if the sampling procedure, which in itself requires a TN contraction, provides an advantage with respect to such a direct calculation. But,

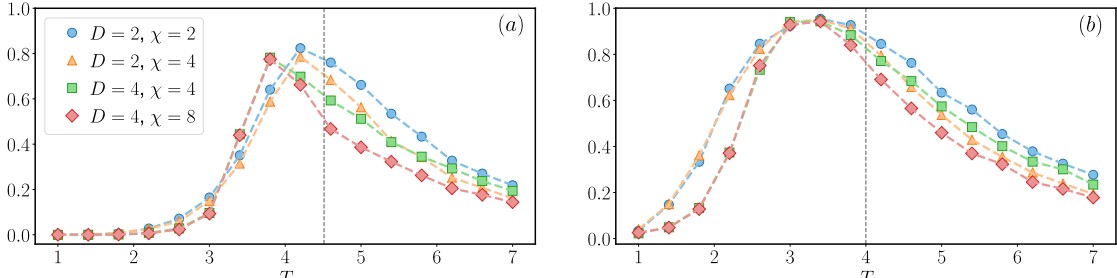

Figure 9: TNMH rejection rates for the three-dimensional Ising model as a function of the temperature. Plot (a) corresponds to a uniform ferromagnet and (b) to a uniform antiferromagnet in a field $h = 3$. $D$ denotes the PEPS bond dimension, while $\chi$ stands for the boundary bond dimension used when compressing the PEPS associated to a plane of the lattice. A lattice of size $16 \times 16 \times 16$ was used, with open boundary conditions, and for each bond dimension and temperature, 50 chains were run for 150 steps each. The critical temperature is $T_c \approx 4.512$ [61] for the ferromagnetic Ising model, and $T_c \approx 4$ [63] for the antiferromagnetic with this field (We attribute the offset with respect to this value to finite size effects.).

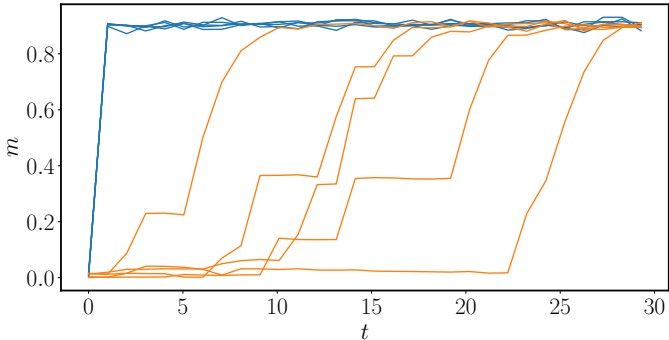

Figure 10: Single site magnetization along different Markov chains at $T = 3$ for two algorithms, Wolff's (orange) and TNMH (blue) ($D = 2, \chi = 2$) for a ferromagnetic $16 \times 16 \times 16$ lattice. $t$ represents the number of cluster moves in the former case and the number of TNMH iterations in the latter.

while the TNMH algorithm can succeed with a very undemanding approximate TN contraction (i.e. using very low bond dimensions), achieving a result of comparable quality by direct contraction generally requires more computational effort. To make this point more concrete, we have compared the value of the average energy in the three-dimensional ferromagnetic case, as obtained with the TNMH scheme and with direct TN contractions with different values of the bond dimensions (Fig. 12). We observe that, at temperatures where the direct contraction with up to $(D, \chi) = (8, 16)$ was not sufficient to obtain an accurate estimate of the energy, the TNMH with $(D, \chi) = (2, 2)$ was successful, since it produced decent acceptance rates, and eventually provided good samples thanks to irreducibility and reversibility.

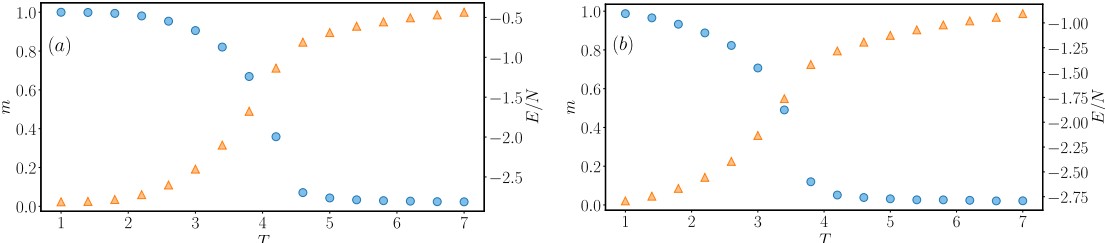

Figure 11: Magnetisation (blue) and energy (orange) per spin of the ferromagnetic Ising model (a) and staggered magnetisation and energy per spin of the antiferromagnetic Ising model in an external field (b) on a cubic $16 \times 16 \times 16$ lattice with open boundary conditions. The error bars are smaller than the symbols. The largest bond dimensions used to obtain the curves were $D = 4, \chi = 8$.

# 5 Other models

In the previous sections we have presented the TNMH algorithm in detail and benchmarked it for the Ising model on two- and three-dimensional square lattices. However, the scheme offers great versatility. In this section, we summarize a number of possibilities to apply and extend the algorithm for more general problems, which will be explored in further detail elsewhere. We show how to deal with arbitrary boundary conditions in Appendix B. The reader interested only in the basic algorithm can safely jump to section 6.

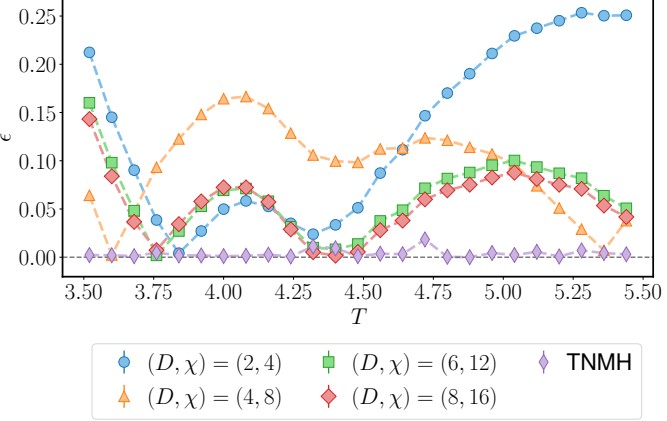

Figure 12: Relative error $\epsilon$ in the average energy per spin computed via different techniques for different temperatures (near the phase transition) for the three-dimensional ferromagnetic Ising model. The reference values are obtained using Wolff's algorithm, purple rhombi are obtained using samples from our algorithm with $(D, \chi) = (2, 2)$, and the other results are obtained taking derivatives of the logarithm of the approximate contraction of the TN representing the partition function.

**The XY model**

In absence of a vector potential, the XY model describes a lattice of planar spins, interacting through the Hamiltonian

$$H_{XY} = -\sum_{\langle i,j \rangle} \cos(\theta_i - \theta_j)\nu, \tag{14}$$

where the local variables are the angles $\{0 \leq \theta_i < 2\pi : i \in V\}$. Although these variables are continuous, this model can be mapped into a system that allows to use a variation of the sampling method used for the Ising model. First, a duality transformation establishes an equivalence between (14) and a system of integer variables residing on the (oriented) links of the lattice involved in four-body interactions [64, 65, 66, 67]. That is, the partition function takes the form

$$Z(\beta) = \lim_{N \to \infty} \prod_{l \in E} \left( \sum_{n_l = -N}^{N} I_{n_l}(\beta) \right) \prod_{i \in V} F_{n_1^{(i)}, n_2^{(i)}}^{n_3^{(i)}, n_4^{(i)}}, \tag{15}$$

where $n_1^{(i)}, n_2^{(i)}, n_3^{(i)}, n_4^{(i)}$ are the values for the links meeting at site $i$. $I_{n_l}(\beta)$ are the modified Bessel functions of the first kind, and

$$F_{n_1, n_2}^{n_3, n_4} = \int_0^{2\pi} \frac{d\theta}{2\pi} e^{i\theta(n_1 + n_2 - n_3 - n_4)} = \delta_K(n_1 + n_2 - n_3 - n_4),$$

where $\delta_K$ denotes the Kronecker delta function.

At fixed $\beta$, $I_{n_l}(\beta)$ decays fast and truncating the sum in Eq. (15) is a sensible approximation. The partition function of the XY model can thus be approximated by a tensor network where the degree of freedom at each bond takes value in a finite set. In the language of Appendix A, the tensor at each site $i$ would now be

$$A_{n_2 n_4 n_1 n_3}^{(i)} = \left( \prod_{k=1}^{4} I_{n_k}(\beta) \right)^{1/2} F_{n_1, n_2}^{n_3, n_4},$$

and the contraction of the TN made up of these tensors would give an approximation $\widetilde{Z}(\beta)$ to the partition function $Z(\beta)$. Similarly, the marginal probability density of the spin at a site $i$, $\widetilde{\pi}^{(\beta)}(\theta_i)$, can be approximated by replacing the tensor at site $i$ with

$$A_{n_2 n_4 n_1 n_3}^{(i)}(\theta_i) = \left( \prod_{k=1}^{4} I_{n_k}(\beta) \right)^{1/2} \frac{e^{i\theta(n_1 + n_2 - n_3 - n_4)}}{2\pi},$$

and normalizing the contraction to the approximate partition function previously obtained. Using renormalisation to approximately contract tensor networks, and the inverse sampling method, a candidate configuration $\omega' = \{\theta_i' : i \in V\}$ can be drawn and accepted or rejected, as we did for Ising models with Algorithm 1. A vector potential could be included [65, 66], and other continuous variable systems admit a similar construction [68].

On top of the bond dimension used for the renormalisation, the number of terms kept in the series expansion of the transfer matrix in Eq. (15) is another parameter that governs the accuracy of the contraction. As for the 3D Ising model discussed above, a tensor network with a low value for this parameter may be accurate enough to sample from and propose moves for a Markov chain, but not precise enough to compute the observables with a single contraction.

A detailed study of the XY model is beyond the scope of this paper. But we have made preliminary computations that show acceptance rates comparable to those of the ferromagnetic Ising model. In order to see how correlated the proposed collective moves are, we have

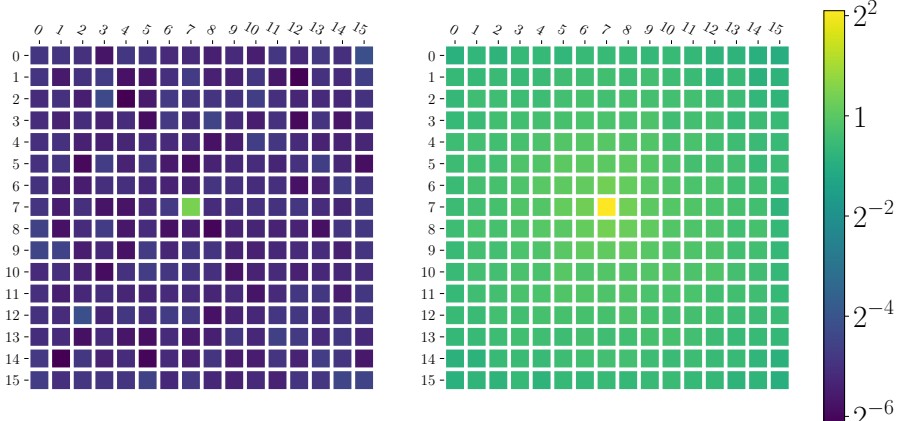

Figure 13: Mutual information in bits between the updates at different sites, that is, in the changes of the angles after a sweep of the Markov chain, $I(\theta_i(t+1) - \theta_i(t) : \theta_j(t+1) - \theta_j(t))$ for two schemes. Left: a local algorithm (a Metropolis single spin flip where the proposed local updates were chosen from $U(0, 2\pi)$). Right: TNMH. The numerical experiment was conducted on a homogeneous XY model with no external field on a 16×16 lattice at a temperature of $T = 0.5$. The bond dimension used in TNMH was $D = 20$, and the number of terms kept in the series of Eq. (15) was $N = 4$, which gave a TN with a local dimension $d = 9$.

computed the mutual information between the updates at different sites of the TNMH algorithm and compared it to that obtained from a local algorithm, Figure 13. The instance chosen for the comparison is the zero-field uniform XY model on a $16 \times 16$ lattice at a temperature $T = 0.5$. The difference in the results is noteworthy, and demonstrates that the TNMH algorithm is indeed capable of producing global correlated updates.

**Drawing configuration differences**

We now show that a TNMH scheme for the Ising model can be extended to deal with other nearest neighbour hamiltonians. For the sake of concreteness, we will focus on the $\lambda\phi^4$ model, defined on a two-dimensional square lattice $\Lambda = (V, E)$ by the energy function

$$H(\{\phi_i\}) = \sum_{\langle i,j \rangle \in E} (\phi_i - \phi_j)^2 + \sum_{i \in V} (\frac{1}{2}m^2\phi_i^2 + \frac{\lambda}{4!}\phi_i^4),$$

where each local variable $\phi_i$ takes value in $\mathbb{R}$. (See also Refs. [32, 69] for the use of tensor networks in lattice field theories.) As usual, we are interested in sampling according to the Boltzmann distribution for some fixed value $\beta$. We will use the following simple lemma.

**Lemma 1** *Any real function of two binary variables, B, can be expressed as an Ising model energy plus some constant:*

$$B(\sigma, \sigma') = J\sigma\sigma' + h\sigma + h'\sigma' + K. \tag{16}$$

*Proof*: (16) defines a system of four linear equations for the four unknowns $J, h, h', K$, one for each assignment $(\sigma, \sigma')$. The determinant of the matrix of this system of equations does not vanish but is equal to 16; a solution to (16) therefore exists for any 4-uple $\{B(\sigma, \sigma')\}$ and is unique.

Let $\omega = \{\phi_i : i \in \Omega\}$ denote the current configuration. A Markov chain with collective updates can be constructed using the TNMH presented for the Ising model in Section 2 if we draw configuration changes. We proceed as follows. An integer $m$ is drawn uniformly and

randomly in $\{0, 1, \ldots, m_{\max}\}$, where $m_{\max}$ is equal to 9, say. $\forall i \in V$, we draw $\gamma_i$ according to a Gaussian distribution with zero mean and variance equal to $10^{-m}$. With $\Gamma = \{\gamma_i : i \in V\}$, we construct the function:

$$H_I(\{\sigma_i\}|\omega, \Gamma) = \sum_{\langle i,j \rangle \in E} (\psi_i(\sigma_i|\phi_i, \gamma_i) - \psi_j(\sigma_j|\phi_j, \gamma_j))^2$$
$$+ \frac{1}{2}\sum_{i \in V} m^2 \psi_i(\sigma_i|\phi, \gamma_i)^2 + \frac{\lambda}{4!}\sum_{i \in V} \psi_i(\sigma_i|\phi_i, \gamma_i)^4,$$

where

$$\psi_i(\sigma_i|\phi_i, \gamma_i) = \frac{1 - \sigma_i}{2}\phi_i + \frac{1 + \sigma_i}{2}(\phi_i + \gamma_i),$$

with $\sigma_i \in \{-1, +1\}$ $\forall i \in V$. By lemma 1, $H_I(\{\sigma_i\}|\omega, \Gamma)$ can be expressed as an Ising Hamiltonian for the variables $\{\sigma_i\}$ (plus some irrelevant global constant):

$$H_I(\{\sigma_i\}|\omega, \Gamma) = -\sum_{i \in V} h_i(\omega, \Gamma)\,\sigma_i - \sum_{\langle i,j \rangle \in \Gamma} J_{i,j}(\omega, \Gamma)\,\sigma_i\sigma_j.$$

The Boltzmann distribution of the Ising model $H_I$,

$$\pi_I^{(\beta)}(\{\sigma_i\}|\{\theta_i\}, \Gamma) = \frac{e^{-\beta H_I(\{\sigma_i\}|\{\theta_i\}, \Gamma)}}{\sum\limits_{\{\sigma_j\}} e^{-\beta H_I(\{\sigma_i\}|\{\theta_i\}, \Gamma)}},$$

can generically not be sampled directly. But we can construct a tensor network approximation $\widetilde{\pi}^{(\beta)}(\cdot|\omega, \Gamma)$ for it, as described in Section 2. Given $\Gamma$ as defined above, let us define $\tau(\Gamma) = \{-\gamma_i : i \in V\}$. The sequence of instructions listed in Algorithm 2 defines an irreducible and reversible Metropolis-Hastings Markov chain that achieves collective updates for the $\lambda\phi^4$ model.

---

**Algorithm 2** Configuration difference collective update

---

1: Draw an integer $m$ u.a.r. in $\{0, \ldots m_{\max}\}$.
2: Draw $|V|$ i.i.d. Gaussians with zero mean and variance equal to $10^{-m}$: $\Gamma = \{\gamma_i : i \in V\}$.
3: Draw $\{\sigma_i : i \in V\}$ according to $\widetilde{\pi}^{(\beta)}(\cdot|\omega, \Gamma)$.
4: Accept the move $\{\phi_i : i \in V\} \to \{\phi_i + \frac{1+\sigma_i}{2}\gamma_i\}$ with probability

$$\min\left\{1, \frac{\widetilde{\pi}_I^{(\beta)}(\{\sigma_i\}|\omega, \Gamma)}{\widetilde{\pi}_I^{(\beta)}(\{\sigma_i\}|\omega', \tau(\Gamma))} \times \frac{\pi^{(\beta)}(\omega')}{\pi^{(\beta)}(\omega)}\right\}.$$

---

The idea of making configuration difference updates appeared in the study of the ferromagnetic XY model, for which the Wolff algorithm for the ferromagnetic Ising model can be recycled [6]. In principle, Algorithm 2 could be applied to frustrated systems.

A class of systems for which we believe it could be useful to draw differences of configurations as described here are matrix models, such as $SU(d)$ lattice gauge theories [70]. The auxiliary Hamiltonian representing the possible choices for a move would no longer be two-body Ising. Still, it is not difficult to construct a tensor network representation for its Boltzmann distribution, as we have done when studying the XY model.

## Triangular lattices

We now show how the construction presented in Section 2, specific to square lattices, can be used as such to deal with a triangular lattice. Let us assume that we are interested in some

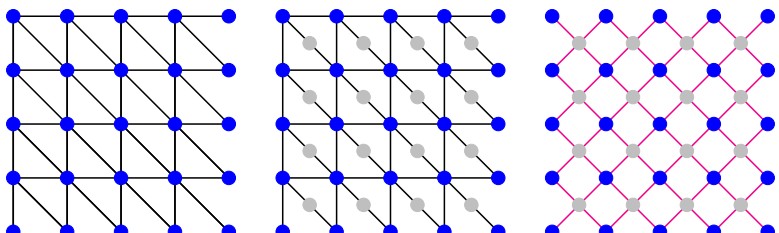

Figure 14: Left: Interaction graph of a triangular lattice system. Centre: Same interaction graph decorated with extra degrees of freedom located on the diagonals (grey dots). Right: Square lattice on which a Hamiltonian $H_\square$ associated with the original system is defined.

particular observable $X$. That is, we wish to estimate

$$\langle X \rangle = \frac{1}{Z(\beta)} \sum_{\omega \in \Omega} X(\omega)\, e^{-\beta H(\omega)} .$$

To this end, we construct an extended model, obtained by decorating the original lattice with extra spins living on each diagonal link as shown on Fig. 14 (a) and (b). With each particle of the original model, we will associate the new extra spin located south east to it. Let $p : V \to V_{\text{new}}$ denote the function that realises this association, where $V_{\text{new}}$ denotes the set of new vertices. The Hamiltonian of the extended model reads

$$H_{\text{ext}}(\omega_{\text{ext}}|\gamma) = H(\omega) - \gamma \sum_{j \in V} \sigma_j \sigma_{p(j)} , \tag{17}$$

for $\gamma > 0$, where $\omega_{\text{ext}} \in \Omega_{\text{ext}} = \{-1, +1\}^{|V \cup V_{\text{new}}|}$. $Z_{\text{ext}}(\beta|\gamma)$ will denote its partition function.

**Proposition 1**

$$\langle X \rangle = \lim_{\gamma \to \infty} \frac{1}{Z_{ext}(\beta|\gamma)} \sum_{\omega_{ext}} X(\omega)\, e^{-\beta H_{ext}(\omega_{ext}|\gamma)} , \tag{18}$$

*whenever $\beta$ and $|V|$ are both finite.*

*Proof:* The extended configuration space $\Omega_{\text{ext}}$ can be decomposed as $\Omega_{\text{ext}} = \Omega_{\text{ext}}^{(0)} \cup \Omega_{\text{ext}}^{(1)} \cup \ldots \cup \Omega_{\text{ext}}^{(|V|)}$, where $\Omega_{\text{ext}}^{(m)}$ denotes the subset of all configurations such that there are exactly $m$ sites $j \in V$ where $\sigma_j \neq \sigma_{p(j)}$. This decomposition induces another for the partition function of the extended model as

$$Z_{\text{ext}}(\beta|\gamma) = \sum_{\omega \in \Omega} e^{-\beta H(\omega) + \beta \gamma |V|} + \sum_{m=1}^{|V|} \zeta_m\, e^{\beta \gamma(|V|-2m)} ,$$

where the coefficients $\zeta_m$ are all finite and independent of $\gamma$. Similarly, the sum appearing in the r.h.s. of (18) can be expressed as

$$\sum_{\omega \in \Omega} X(\omega) e^{-\beta H(\omega) + \beta \gamma |V|} + \sum_{m=1}^{|V|} \xi_m\, e^{\beta \gamma(|V|-2m)} ,$$

where the coefficients $\xi_m$ are also finite and independent of $\gamma$. Finally, it is obvious that the ratio

$$\frac{\sum_{\omega \in \Omega} X(\omega) e^{-\beta H(\omega) + \beta \gamma |V|} + \sum_{m=1}^{|V|} \xi_m\, e^{\beta \gamma(|V|-2m)}}{\sum_{\omega \in \Omega} e^{-\beta H(\omega) + \beta \gamma |V|} + \sum_{m=1}^{|V|} \zeta_m\, e^{\beta \gamma(|V|-2m)}} ,$$

tends to $\langle X \rangle$ in the limit where $\gamma$ tends to infinity.

A similar argument provides the following identity between the Boltzmann weight for a configuration of the extended space $\omega_{\text{ext}}$ and the Boltzmann weight of its restriction to $\Omega$, $\omega$:

$$\lim_{\gamma \to \infty} \frac{e^{-\beta H_{\text{ext}}(\omega_{\text{ext}}|\gamma)}}{Z_{\text{ext}}(\beta|\gamma)} = \prod_{j \in V} \delta_{\text{K}}(\sigma_j, \sigma_{p(j)}) \frac{e^{-\beta H(\omega)}}{Z(\beta)}. \tag{19}$$

The contribution of any site $j$ of the original lattice $\Lambda$ to the numerator of the r.h.s. of (19) reads

$$\delta_{\text{K}}(\sigma_j, \sigma_{p(j)}) \exp\Big( \beta \Big( h_j \sigma_j + \sum_{k \in N(j)} J_{jk} \sigma_j \sigma_k \Big) \Big), \tag{20}$$

where $N(j)$ denotes the neighbourhood of $j$. Because of the Kronecker delta, for any bipartition of this neighbourhood $N(j) = N'(j) \cup N''(j)$, (20) remains invariant if the sum in the exponential is substituted with

$$\sum_{k \in N'(j)} J_{jk} \sigma_{p(j)} \sigma_k + \sum_{k \in N''(j)} J_{jk} \sigma_j \sigma_k. \tag{21}$$

Assuming w.l.o.g. the boundary conditions represented on Fig. 14-left, we choose, for every site $j$, $N'(j)$ to consist in the sites located east, south, and south east of $j$, $\forall j \in \Lambda$. (Edge and corner sites might require different choices of subsets $N'(j)$, depending on the boundary conditions.) This choice results in a *square* lattice hamiltonian $H_{\square}$ whose couplings are shown on Fig. 15, and whose interaction graph is displayed on Fig. 14(c).

Let $\widetilde{\pi}_{\square}$ denote a probability distribution approximating the Boltzmann distribution associated with $H_{\square}$ through tensor network renormalisation. To deal with a triangular lattice using a TNMH code for a square lattice, a possibility is a Markov chain where, at each step, a candidate configuration $\omega'_{\text{ext}}$ is drawn according to $\widetilde{\pi}_{\square}$, and the move from the current configuration $\omega_{\text{ext}}$ to this candidate is accepted with Metropolis-Hastings probability:

$$\min \left\{ 1, \frac{e^{-\beta H(\omega')}}{e^{-\beta H(\omega)}} \times \frac{\widetilde{\pi}_{\square}(\omega_{\text{ext}})}{\widetilde{\pi}_{\square}(\omega'_{\text{ext}})} \right\},$$

where $\omega$ (resp. $\omega'$) denotes the restriction of $\omega_{\text{ext}}$ (resp. $\omega'_{\text{ext}}$) to $\Omega$.

This mapping from a triangular lattice to a square lattice doubles the number of sites but we stress that the bond dimension of the (square) tensor network associated is unchanged and equal to that of the local degrees of freedom ($d = 2$ for the Ising model). It would be very interesting to see whether the argument can be extended to three dimensions, and for example map a body centred cubic lattice model to a simple cubic lattice model.

A *quantum* analogue of the mapping exists: square PEPS can be used for a triangular quantum spin Hamiltonian. The extended Hamiltonian (operator) now reads $H_{\text{ext}} = H - \gamma \sum_{j \in V} \sigma_j^z \sigma_{p(j)}^z$. Proposition 1 still holds true if $\sum_{\omega_{\text{ext}}} X(\omega)\, e^{-\beta H_{\text{ext}}(\omega_{\text{ext}}|\gamma)}$ is substituted with $\text{Tr}\, X e^{-\beta H_{\text{ext}}}$. Expressing the trace in the basis of eigenstates of $\{\sigma_j^z\}$ operators, an analogue of the substitutions (20,21) holds true too. If for example, one wants a TNS approximation of the ground state, one could alternate Trotter steps with applications of the projector $|00\rangle\langle00|_z + |11\rangle\langle11|_z$ on each particle of the original lattice and its partner. Actually, a further reduction can be made: one readily checks that the interaction graph transformation shown on Fig. 14 produces a hexagonal lattice when applied to a square lattice. Therefore, in principle, it should even be possible to study triangular lattices with hexagonal PEPS.

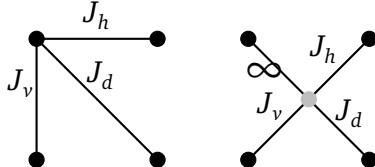

Figure 15: Couplings in and around a plaquette in the original and extended models (left and right respectively). The new couplings produce a square lattice rotated by a $\pi/4$ angle with respect to the original lattice.

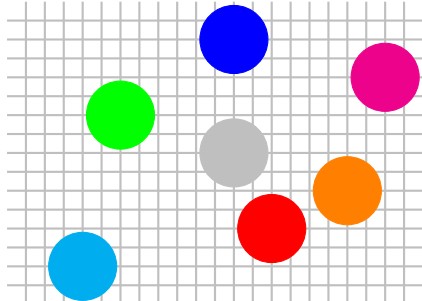

Figure 16: Example of a configuration of hard disks in a discretised volume. (Periodic boundary conditions assumed.)

## Hard spheres

To close this section, we show how tensor network contractions can also be used to implement collective Monte Carlo updates for systems of hard spheres (or disks in two dimensions) [5]. We will combine three ideas for that purpose. The first is a discretisation of the domain that contains the spheres. The second is a shift of perspective where a configuration will not so much be regarded as a collection of locations for the spheres, but rather as the specification for the state of each cell of the volume that contains them (occupied or empty). The third is to use a tensor network to encode possible changes for each cell.

We consider a system of $N$ hard disks in two dimensions confined in a square area discretised with a square lattice ($M$ cells). Although this is not essential, we will assume periodic boundary conditions in order to keep the presentation simple. $N$ is fixed, as well as the lattice spacing $\epsilon$. All disks have identical radius. A configuration is said to be *valid* if (i) the centre of each disk is pinned on the intersection of a vertical and a horizontal line of the lattice, (ii) no cell contains bits of matter belonging to different disks. Fig. 16 is an example of a valid configuration.

Our goal is to sample uniformly amongst all valid configurations. For that, we will design a Markov chain of collective updates where each disk either stands still or is moved vertically or horizontally by one lattice spacing. A configuration change must comply with the following rules:

1. A disk cannot be split.

2. A disk cannot be compressed.

3. Disks cannot overlap, not even completely (conservation of particle number).

We will assume the disks are distinguishable and we will associate a label $\{1, 2, \ldots, N\}$ to each of them, which is why each disk appears with a different colour in the illustration of



Figure 17: Diagrammatic representation of the PEPS tensor associated with each cell $j$ of the lattice.

Fig. 16. We will denote $S_0$ the set of empty cells, and $S_\alpha$ the set of cells occupied by disk $\alpha$, $1 \leq \alpha \leq N$.

*Given* a valid configuration $\omega$, we associate a tensor $P_j$ with each cell $j$ of the lattice, see Fig. 17. The index $\sigma$ of this tensor encodes the move that a bit of matter located at cell $j$ would undergo: $\mathcal{M} \equiv \{0, -1, +1, -2, +2\}$ for {stillness, displacement to the left, displacement to the right, downwards displacement, upwards displacement} respectively. The role of the $w, e, n, s$ degrees of freedom of $P_j$ is to communicate the chosen move at $j$ to its neighbour cells; the indices $w', e', n', s'$ provide the information about the moves made in neighbouring cells to cell $j$. We want to assign values to these tensors $\{P_j\}$ that guarantee moves can only occur between valid configurations.

A. **Initialisation.** For each cell $j$, $P_j(\sigma)_{w,e,n,s}^{w',e',n',s'} = 1 \ \forall \sigma, w', e', n', s', w, e, n, s \in \mathcal{M}$.

B. **Empty cells.** $\forall j \in S_0$, since there is no matter to be moved, we decree that $P_j(\sigma)_{w,e,n,s}^{w',e',n',s'} = 0$, $\forall w', e', n', s', w, e, n, s$ if $\sigma \neq 0$ (holes do not move).

C. **Faithful move communication.** $\forall j$, $P_j(\sigma)_{w,e,n,s}^{w',e',n',s'} = 0$ unless $w = e = n = s = \sigma$.

D. **Rigidity.** Let $j, k$ denote two neighbouring cells covered by a same disk $S_\alpha, \alpha \neq 0$. Let us assume, say, that $j$ is located left to $k$. We impose that $P_j(\sigma)_{w,e,n,s}^{w',e',n',s'} = 0$ if $e \neq w'$. Similar constraints are imposed on all other pairs of cells $j, k$ covered by a same disk and such that $|j - k| = 1$.

E. **Prevention of collisions.** By definition, a collision has occurred between two disks $\alpha$ and $\alpha'$ if and only if two bits of matter belonging to $\alpha$ and $\alpha'$ respectively are found in a same cell. Therefore, it is necessary and sufficient to forbid all such events in order to prevent a collision. If there is a collision, either one disk is immobile, say $\alpha$, and $\alpha'$ moves by one cell to overlap with $\alpha$ (case A), or both $\alpha$ and $\alpha'$ move to cause the overlap (case B).

Case A occurs if and only if there are pairs of adjacent cells $c$ and $c'$ in $S_\alpha$ and $S_{\alpha'}$ respectively which content will occupy a same cell. To prevent the collision, it is sufficient to impose that for each such pair $(c, c')$, the bit of matter contained in $c'$ cannot hop in $c$. There are four such moves to prohibit; they are represented by the four leftmost drawings of Fig. 18.

In case B, $\alpha$ and $\alpha'$ either move along a same direction (case BI) or along perpendicular directions (case BII). Case BI occurs if and only if there are pairs of cells $c$ and $c'$, separated by one cell, in $S_\alpha$ and $S_{\alpha'}$ respectively, which contents are moved closer to each other along a common line. It is thus enough to prevent the events represented by the rightmost drawings of Fig. 18. Case BII is dealt with similarly, and results in the prohibition of the events represented by the four remaining diagrams of Fig. 18.

Collisions where a bit of matter contained in a cell $k \in S_{\alpha'}$ moves to its left, and lands in a cell $j \in S_\alpha$ already occupied by a bit of matter that does not change its position, can be prevented by imposing $P_j(0)_{0000}^{-1,e',n',s'} = 0 \ \forall e', n', s'$. The other A prohibitions admit similar translations into constraints on the tensors, and the six B prohibitions can be enforced likewise. For example the prohibition of the move depicted on the diagram located rightmost top of Fig. 18 translates into $P_j(\sigma)_{w,e,n,s}^{+1,-1,n',s'} = 0 \ \forall w, e, n, s, n', s'$ whenever the left and right

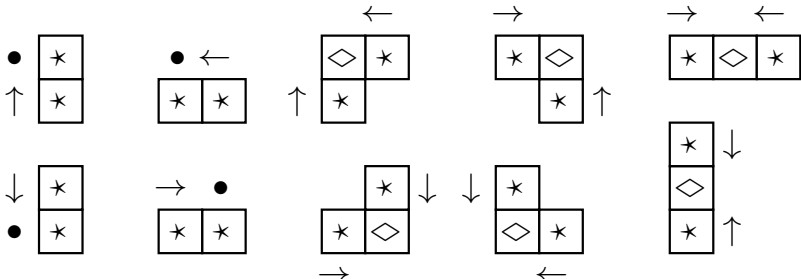

Figure 18: Forbidden moves in the discretised hard disks model. An asterisk in a cell indicate presence of matter, the rhombus symbol stands for a cell that can either be empty or filled. A dot on the side of a cell indicates no move, whereas an arrow indicates a move by one lattice spacing and its direction.

neighbours of cell $j$ are occupied by different disks.

The exact contraction of all tensors yields a function $Q(\sigma_1, \ldots, \sigma_M | \omega)$, which value is equal to 0 if the move $\{\sigma_1, \ldots, \sigma_M\}$ is forbidden from configuration $\omega$, and 1 otherwise. We note that for a fixed assignment $\{\sigma_1, \ldots, \sigma_M\}$, $Q(\sigma_1, \ldots, \sigma_M | \omega)$ can be evaluated exactly. Ideally, we would construct a Metropolis-Hastings Markov chain where the moves are sampled according to the prior

$$\pi_{\text{id}}(\sigma_1, \ldots, \sigma_M | \omega) = \frac{Q(\sigma_1, \ldots, \sigma_M | \omega)}{\sum_{\{\tau\}} Q(\tau_1, \ldots, \tau_M | \omega)}.$$

As we don't expect this to be possible, we propose to approximate $\pi_{\text{id}}$ through tensor network renormalisation, as we did for Ising models. At fixed volume $M$, the computational cost for constructing the tensors scales as $1/\epsilon^2$. The bond dimension of the tensor network is *independent* of $\epsilon$. As for Ising models, acceptance rates should increase with the bond dimension used in the tensor network renormalisation. If necessary, a complementary strategy to increase acceptance rates is to select a region at each Markov step, and impose that all disks outside of it or touching its boundary remain fixed; such a region would vary from one time step to the next and may even be disconnected.

In two dimensions, the hard sphere model is known to exhibit a fluid-solid phase transition for a filling fraction $\eta = \pi a^2 N / A \simeq 0.7$, where $a$ is the radius of the disks, and $A$ denotes the area of the domain that contains them [5] ($A = M\epsilon^2$ here). It would be very interesting to see how a finite value of $\epsilon$ affects this phase transition. Actually, because of the discretisation, the model considered here is, strictly speaking, not the hard sphere model discussed in [5], for which the disks could in principle occupy any position in Euclidean space. It might be that the phase transition in the limit $\epsilon \to 0$ does not correspond to the transition point of the hard sphere model defined in Euclidean space. But it just might if a different lattice geometry is used. A similar phenomenon occurs in the study of fluids with cellular automata: square lattices do not relate to the Navier-Stokes equation whereas triangular lattices do [71].

A construction similar to Fig. 17 should hold for hard spheres in three dimensions, and we believe that an analogue also exists for dimer (and dimer-monomer) models. In this latter case, the possibility to rotate dimers by a $\pi/2$ angle produces additional constraints on the tensors.

# 6 Discussion

The interplay between Monte Carlo and tensor network methods is a rich and vastly unexplored subject. While various previous works have reported on using Monte Carlo sampling for tensor network contractions, we have here presented an analysis of the converse: a new class of Markov chain Monte Carlo algorithms for many-body classical systems based on tensor network renormalisation. This class belongs in the family of Metropolis-Hastings schemes. Our construction produces collective updates. It is also irreducible and reversible; as such, asymptotic convergence towards the target probability distribution is guaranteed. We emphasize its universal nature: it works the same for any nearest neighbour Hamiltonian with finite local degrees of freedom.

We have benchmarked our scheme for a variety of instances of the two-dimensional Ising model defined on a square lattice. For ferromagnets and antiferromagnets, very high acceptance rates have been observed for larger systems, even with modest values of the bond dimension. Besides, drops in acceptance rates have been shown to signal criticality. Looking at equilibration and decorrelation times, the scheme compares extremely well with single spin flip updates and Wolff algorithm. As expected, the scheme's performance is lower for frustrated and disordered instances than for the ferro- and antiferromagnets. Still, our results are very encouraging. In particular, for disordered instances, equilibration appears to be occurring orders of magnitude faster than for state-of-the-art techniques such as parallel tempering supplemented with isoenergetic cluster moves, both when time is counted in Monte Carlo steps and in seconds.

We have also demonstrated the potential of the method for three dimensional systems, by testing it on ferromagnetic and antiferromagnetic instances. Also in this case, we have observed faster equilibration as compared to Wolff algorithm and, remarkably, even with a simple contraction strategy and small bond dimension, the scheme can be shown to remain usable at near critical temperatures, whereas a much more costly direct TN contraction results in considerable errors.

We have used simple procedures to implement tensor network renormalisation, and we have made no particular effort to write an efficient code. For these reasons, we believe the results presented here could be substantially improved. It would also be very interesting to study what can be gained by using other renormalisation schemes for approximate contractions of tensor networks [19]. For example, schemes involving disentanglers would be a natural option in this regard [22]. Also for future work is the study of how TNMH Markov chains combine with parallel tempering [72].

A major advantage of our construction is its versatility. We have seen that with little extra effort, a code valid for the Ising model on a square lattice can be used as such to construct a collective update Markov chain in other settings such as the XY model, or a triangular lattice, and that TNMH could also be used to study gases of hard spheres. In principle lattice systems with long range interactions could also be considered. For instance, given an Ising Hamiltonian $H$ where the interactions decay with the distance as a power law, one can associate an auxiliary Hamiltonian $H_{\varrho}$ where all interactions within some range $\varrho$ are identical to $H$, and all interactions beyond $\varrho$ have been truncated. One can next construct a tensor network prior from this Hamiltonian $H_{\varrho}$. Two parameters would now govern the Markov chain: the bond dimension and the range $\varrho$. We have also restricted ourselves to scalar degrees of freedom in this work. But the discussion held in Section 5 shows that TNMH sampling should also apply to matrix models, in particular lattice gauge theories.

A natural variation of our work would be to depart from tensor network representations and use a quantum device to prepare Gibbs states and estimate the probability to draw a given configuration [73, 74]. Such a device would be called as an external subroutine in

(classical) Metropolis-Hastings iterations. Just as our 3D computations have revealed that inaccurate contraction schemes could still be useful for sampling, it would be very interesting to investigate how much computational power such quantum devices retain when imperfect. These ideas will be studied elsewhere.

Finally, it would be instructive to develop a mathematical perspective on the schemes presented here. In particular, we believe it would be meaningful to identify a non-trivial model for which the mixing time associated with our TNMH scheme could be upper bounded, *e.g.* using a log-Sobolev inequality [75]. It would be insightful to establish the dependence of the log Sobolev constant with the bond dimension.

# Acknowledgements

We thank J.I. Cirac and F. Verstraete for fruitful discussions. This work was partly supported by the Deutsche Forschungsgemeinschaft (DFG, German Research Foundation) under Germany's Excellence Strategy – EXC-2111 – 390814868, and by the European Union through the ERC grant GAPS (Grant no. 648913), by Ministerio de Ciencia, Innovación y Universidades (Spain) (grant no. PGC2018-095862-B-C21, 'Tecnologías cuánticas teóricas', grant no. PID2020-113523GB-I00, 'Análisis Matemático y Teoría de Información Cuántica', grant no. MTM2017-88385-P, grant no. SEV-2015-0554, and grant no. CEX2019-000918-M, 'Maríaa de Maeztu'), by Generalitat de Catalunya (Spain), SGR 1761, and from the European Union Regional Development Fund within the ERDF Operational Program of Catalunya (Spain) (project QUASICAT/QuantumCat, ref. 001- P-001644) and by Comunidad de Madrid (Spain) (grant QUITEMAD-CM, ref. S2018/TCS-4342).

# A  MPS renormalisation

We here review the relation between tensor networks and partition function [13, 14, 15, 19, 21]. The setup is a slight generalization of that of Section 2. That is, we consider a nearest neighbour classical Hamiltonian

$$H(\omega) = \sum_{\langle i,j \rangle} \varphi_{ij}(\sigma_i, \sigma_j),$$

on a lattice $\Lambda = (V, E)$, where the local variables $\sigma_i$ now take value in any finite set, which size we are going to denote $d$. For the sake of simplicity, and without loss of generality, we will again only consider squares lattices, and first limit ourselves to two-dimensional systems for now. At fixed inverse temperature $\beta$, the partition function can be expressed as

$$Z(\beta) = \sum_{\omega \in \Omega} \prod_{\langle i,j \rangle \in E} W_{ij}(\sigma_i, \sigma_j), \tag{A.1}$$

where $W_{ij}$ is a $d \times d$ matrix, whose entries represent all possible contributions of the bond $\langle i,j \rangle$ to the Boltzmann weight of the model, i.e. $W_{ij}(\sigma, \sigma') = e^{-\beta \varphi_{ij}(\sigma,\sigma')}$. As an example, for the Ising model without external magnetic field, the energy associated with a given bond $\langle i,j \rangle$ reads $\varphi_{ij}(\sigma, \sigma') = -J_{ij}\sigma\sigma'$, and the $2 \times 2$ matrix $W_{ij}$ is

$$W_{ij} = \begin{pmatrix} e^{\beta J_{ij}} & e^{-\beta J_{ij}} \\ e^{-\beta J_{ij}} & e^{\beta J_{ij}} \end{pmatrix}. \tag{A.2}$$

We will use the diagrammatical notation in which a tensor is represented by a vertex or a small shape, with as many legs sticking out as there are indices; and where joining two lines

represents a contraction of the corresponding indices. For example, a matrix $W_{ij}$ is represented as follows,

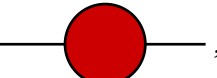

$Z(\beta)$ can be expressed as a tensor network if we shift from a description in terms of matrices associated with the bonds of the lattice (A.1) to a description in terms of tensors associated with its vertices. Let us consider some vertex $i$ with four neighbours and let $e(i)$ denote the vertex to its right. We decompose $W_{i,e(i)}$ as:

$$W_{i,e(i)}(\sigma, \sigma') = \sum_{\mu=1}^{d} L_i(\sigma, \mu) R_{e(i)}(\mu, \sigma').$$

Graphically,

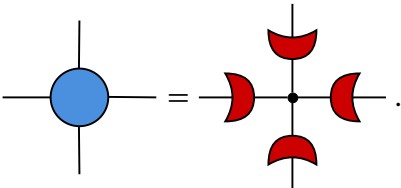

This can be achieved e.g. through a singular value decomposition (SVD) $W_{i,e(i)} = U_{i,e(i)} \Sigma_{i,e(i)} \times V^{\dagger}_{i,e(i)}$, and by setting $L_i = U_{i,e(i)} \sqrt{\Sigma}_{i,e(i)}$, $R_{e(i)} = \sqrt{\Sigma}_{i,e(i)} V^{\dagger}_{i,e(i)}$. Similarly, if $n(i), w(i), s(i)$ denote vertices located above, to the left, and below $i$ respectively, three additional SVD provide the decompositions

$$W_{w(i),i}(\sigma, \sigma') = \sum_{\nu=1}^{d} L_{w(i)}(\sigma, \nu) R_i(\nu, \sigma'),$$

$$W_{i,n(i)}(\sigma, \sigma') = \sum_{\rho=1}^{d} B_i(\sigma, \rho) T_{n(i)}(\rho, \sigma'),$$

$$W_{s(i),i}(\sigma, \sigma') = \sum_{\tau=1}^{d} B_{s(i)}(\sigma, \tau) T_i(\tau, \sigma').$$

We associate a 4-index tensor $A^{(i)}(\sigma)$ with each site $i$ having four neighbours and each spin value $\sigma$, whose components are

$$A^{(i)}_{\mu\nu\rho\tau} = \sum_{\sigma=1}^{d} L_i(\sigma, \mu) R_i(\nu, \sigma) B_i(\sigma, \rho) T_i(\tau, \sigma). \tag{A.3}$$

In diagrammatic notation, Eq. A.3 reads

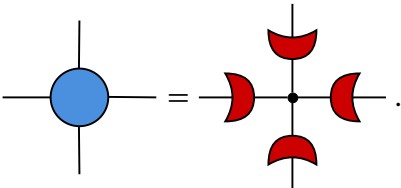

For a system with open boundary conditions, vertices with only three or two neighbours are dealt with likewise. With these tensors, the partition function can be expressed as

$$Z(\beta) = \mathcal{C}(\{A^{(i)}\}), \tag{A.4}$$

where $\mathcal{C}(\{A^{(i)}\})$ denotes the contraction of all the tensors associated with all sites. The entire process from (A.1) to (A.4) is illustrated on Figure 19 for a $4 \times 4$ lattice.

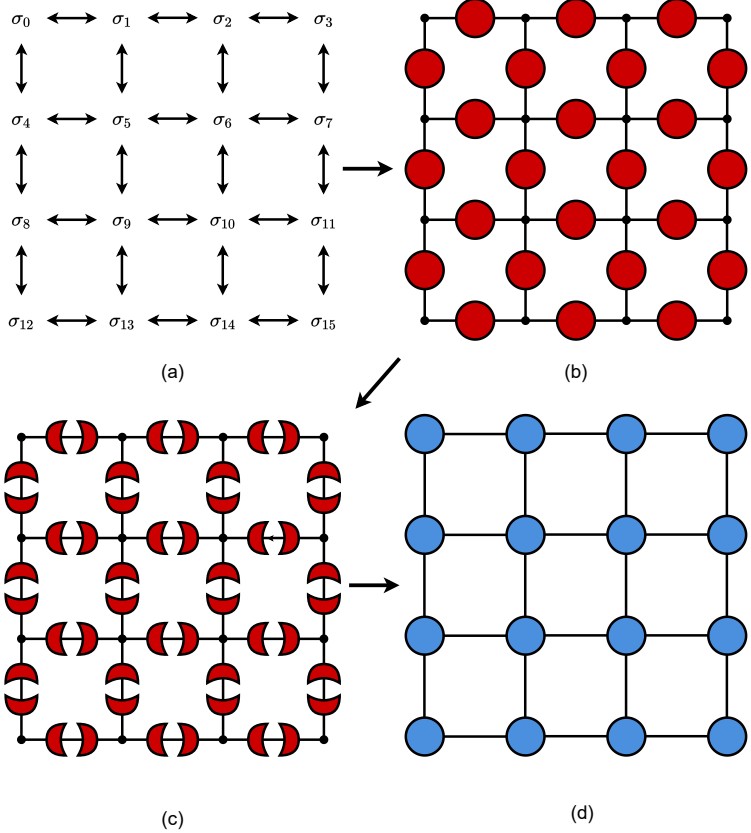

Figure 19: Graphical depiction of the construction of the TN associated with the partition function of a nearest neighbour classical Hamiltonian (4 × 4 lattice in this illustration). (a) We start with a labelling of the vertices of the lattice in consideration. (b) Diagrammatic representation of the Boltzmann weights (red circles) associated with each edge; their contraction yields the partition function. (c) and (d) Singular value decomposition of each $W$ matrix, and regrouping into tensors associated with each vertex of the lattice.

Similarly, one can construct a TN representation of the partition function with some fixed value $s$ for the degree of freedom at site $i$, $Z(\beta|\sigma_i = s)$. It is for instance sufficient that for each neighbour of $i$, $j$, we replace $W_{i,j}(\sigma, \sigma')$ with $W_{ij}^{(s)}(\sigma, \sigma') = \delta_{s,\sigma} W_{i,j}(\sigma, \sigma')$. The ratio of the two quantities, $Z(\beta|\sigma_i = s)/Z(\beta)$, would exactly be the marginal probability $\pi_i^{(\beta)}(s)$ of that spin being in state black $s$, that is, a ratio of two TN contractions. Similarly, one can express any conditional probability $\pi_k^{(\beta)}(\sigma_k|s_1 \ldots s_{k-1})$ as a ratio of two TN contractions:

$$\pi_k^{(\beta)}(\sigma_k|s_1 \ldots s_{k-1}) = \frac{Z(\beta|s_1 \ldots s_{k-1}\sigma_k)}{Z(\beta|s_1 \ldots s_{k-1})}. \tag{A.5}$$

As explained in Section 2, if one were able to evaluate TN contractions exactly, one would have a means to sample according to the Boltzmann distribution exactly. In general, it is only possible to compute approximations to the contractions appearing in the ratio (A.5) and as a result, get an approximation $\tilde{\pi}_k^{(\beta)}$ to $\pi_k^{(\beta)}$. Instead of using these approximations for direct sampling with systematic errors, one can use them as a prior for a reversible Metropolis-Hastings Markov chain. The impossiblity to carry out exact TN contraction then translates into more controllable statistical errors.

For the approximate contraction of an $L \times L$ lattice, we have used one of the simplest schemes available [15, 10]. We define |top⟩ to be the tensor resulting from contracting all

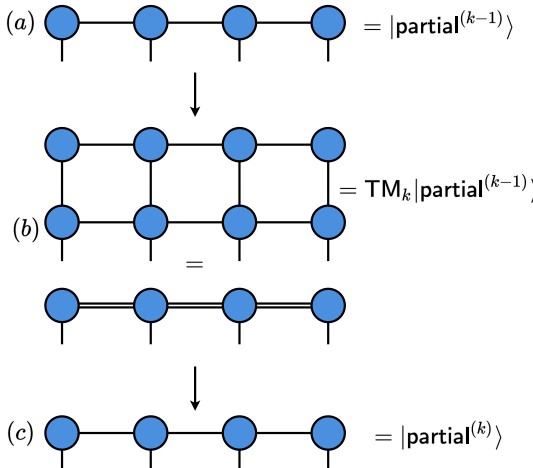

Figure 20: Graphical depiction of the process of contracting a two-dimensional lattice. (a) Approximation to the contraction of the $k-1$ top most rows, $|\text{partial}^{(k-1)}\rangle$. (b) The approximation to $|\text{partial}^{(k)}\rangle$ is constructed by applying the MPO associated with row $k$, $\text{TM}_k$, on the MPS obtained by approximate contraction of the $k-1$ first rows, $|\text{partial}^{(k-1)}\rangle$. The result of that MPO-MPS multiplication is a MPS with a larger bond dimension. (c) Using standard MPS techniques, the product can be approximated by a MPS with lower bond dimension.

the top row tensors along horizontal edges; the remaining free indices after this contraction are legs pointing downward. Similarly, we will call transfer matrix the tensor resulting from a contraction of the tensors along a horizontal bulk row; the transfer matrix resulting from contracting the tensors of row $k$ will be denoted $\text{TM}_k$, $2 < k < L$. Finally, in analogy to $|\text{top}\rangle$, we will denote $|\text{bot}\rangle$ the contraction of bottommost tensors. With these notations, the partition function can be expressed as

$$Z(\beta) = \langle \text{bot}|\text{TM}_{L-1}\dots\text{TM}_2|\text{top}\rangle. \tag{A.6}$$

Both $|\text{top}\rangle$ and $\langle\text{bot}|$ are matrix product states (MPS), whereas the transfer matrices $\text{TM}_k$ are matrix product operators (MPO), all with a bond dimension and a 'physical' dimension equal to $d$; their length is equal to $L$. Our approximation of $Z(\beta)$ is obtained by estimating the rhs of (A.6) sequentially. We initialise $|\text{partial}^{(1)}\rangle \equiv |\text{top}\rangle$, and for $k \in \{2\dots L-1\}$, we define $|\text{partial}^{(k)}\rangle$ to be an MPS approximation to $\text{TM}_k|\text{partial}^{(k-1)}\rangle$ obtained by matrix product state renormalisation, see Fig. 20. $Z(\beta)$ is finally approximated with $\langle\text{bot}|\text{partial}^{(L-1)}\rangle$. The cutoff parameter $D$ sets the accuracy of the approximation. There are many methods available for the renormalisation. Throughout this work, we have mostly used the scheme based on successive SVD [11]. Two-site variational compression has been used to explore equilibration of the two dimensional Ising model with Gaussian disorder [11].

The same method allows to approximate the partition function of a system where some spins have been set to definite values, $\widetilde{Z}(\beta|\sigma_1\dots\sigma_k)$. The only difference is that for such a site $i$ with spin value $\sigma_i$, the tensor (A.3) is replaced with

$$L_i(\sigma_i,\mu)R_i(\nu,\sigma_i)B_i(\sigma_i,\rho)T_i(\tau,\sigma_i). \tag{A.7}$$

As claimed in section 2, an approximate Boltzmann weight $\widetilde{\pi}(\omega)$ can be evaluated since, using Bayes theorem, this probability can be expressed as

$$\frac{\widetilde{Z}(\beta|\sigma_1)}{\widetilde{Z}(\beta)} \times \dots \times \frac{\widetilde{Z}(\beta|\sigma_1\dots\sigma_n)}{\widetilde{Z}(\beta|\sigma_1\dots\sigma_{n-1})}.$$

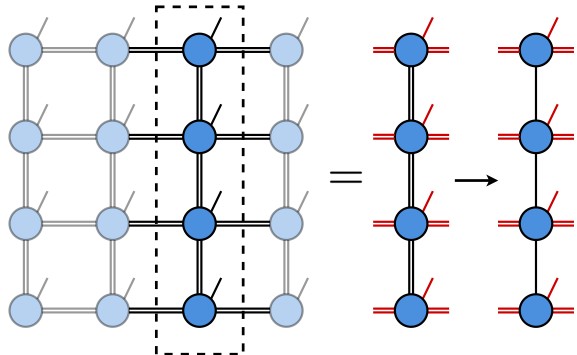

Figure 21: Renormalisation of a black column of a PEPS. For the truncation of the bond dimension of a given a column, its tensors are singled out. The physical bond dimension and the bond dimension that connects these tensors to other columns, drawn in red in this diagram, are treated as the physical dimension of an auxiliary MPS. The bond dimension of that MPS is reduced using a standard truncation algorithm. The resulting tensors of the obtained MPS with lower bond dimension are inserted back in the PEPS. While this procedure has no guarantee of optimality, it is computationally cheap and works well in practice.

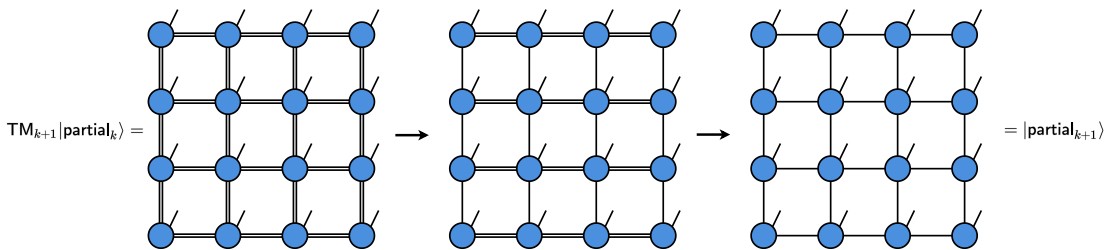

Figure 22: Renormalisation of a PEPS used to apply the TNMH algorithm to three-dimensional systems. The bond dimension is first reduced along horizontal bonds, next along vertical bonds.

Two remarks are in order. First, if the tensors $A^{(i)}$ are well conditioned and if $D$ is high enough, the approximations to partition functions we construct will be strictly positive. So will then be the approximated probabilities (7), and the TNMH is irreducible. Second, if the tensors $\{A^{(i)}\}$, the MPS $|\text{top}\rangle$, $|\text{bot}\rangle$ and the transfer matrices $\text{TM}_k$ are stored, a TNMH update of the whole lattice can be performed at a computational cost that scales linearly with the lattice size.

Plaquette interactions can be dealt with similarly. Using singular value decomposition for the Boltzmann weights and regrouping all the matrices relating to a given site, one obtains a ($\pi/4$ rotated) square lattice for the partition function. Bayes formula can thus again be used for sampling.

We have dealt with three-dimensional models in a similar fashion. Assuming an $L \times L \times L$ lattice, the identity (A.4) can again be obtained after sequence of SVD; $A^{(i)}$ is now a six-leg tensor (for a bulk spin). (A.6) is also still valid, but $|\text{top}\rangle$, $|\text{bot}\rangle$ are now projected entangled pair states (PEPS) instead of MPS, and the transfer matrices $\text{TM}_k$ projected entangled pair operators (PEPOs) instead of MPOs. Just as in two dimensions, without any cutoff, the bond dimension of $\text{TM}_{L-1} \ldots \text{TM}_2|\text{top}\rangle$ would generically grow exponentially with $L$, and renormalisation is in order. There exists a plethora of methods to contract three dimensional tensor

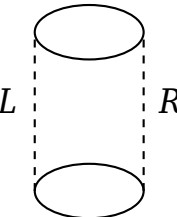

Figure 23: Cylindrical boundary conditions obtained by identification from a rectangle. *L* and *R* denote lines of spins alternating frozen in order to be able to use a sampling scheme designed for open boundary conditions.

networks [76, 21]. We have not aimed at optimality and have opted for simplicity. Again, denoting $|\text{partial}_k\rangle$ the approximate contraction of the first $k$ layers of the TN, the core of the renormalisation consists in constructing a PEPS approximation $|\text{partial}_{k+1}\rangle$ for the contraction $\mathsf{TM}_{k+1}|\text{partial}_k\rangle$. When a PEPO is superimposed on a PEPS, the resulting state is a PEPS with a larger bond dimension.

The bond dimension of $\mathsf{TM}_{k+1}|\text{partial}_k\rangle$ has been reduced by recycling the technique used to compress the bond dimension of MPS. First, an index reshuffling allows us to regard each column of the PEPS $\mathsf{TM}_{k+1}|\text{partial}_k\rangle$ as an MPS, with an effective physical index at each site given by lumping the original physical index of the PEPS with the horizontal virtual indices at that site. The virtual bonds of that MPS are the vertical virtual bonds of the corresponding column of the PEPO. These 'thick' bonds are compressed (or renormalised) as before. This compression along columns is illustrated on Fig. 21. The resulting tensors from the compression are then inserted back into the PEPS, and the same compression is next performed on the horizontal bonds of the PEPS. See Fig. 22 for a depiction of this PEPS renormalisation.Two parameters now govern the accuracy of the approximation: the bond dimension of the PEPS $|\text{partial}_k\rangle$, $D$, and the cutoff for the approximate contraction of two rows of a PEPS, $\chi$ [76].

# B   Arbitrary boundary conditions

Although we have focused on systems with open boundary conditions, the Markov chain (4) allows us to deal with any topology that can be obtained from a rectangle by appropriate identifications. Let us show how with the simple example of a cylinder. If we make an update where we decide to leave a column of spins unchanged, *e.g.* the dashed column '*L*' of Fig. 23, we will effectively be considering a model with open boundary conditions, where the spins in the neighbourhood of the frozen line are subjected to a local extra magnetic field. Such a model can be sampled as before. In order to make sure all spins are refreshed, the cut of frozen spins alternates between the opposite lines depicted as '*L*' and '*R*' respectively. Alternatively, the lines of spins where we choose to cut our system can be chosen randomly.

We have implemented this adaptation of an OBC TNMH code in order to study the equilibration of the two dimensional gaussian spin glass studied in the case of periodic boundary conditions. This has allowed us to compare directly our results with the state-of-the-art results of [56]. Our findings are summarized in Table 2.

As can be appreciated, this adaptation of Algorithm 1 yields equilibration in a number of steps significantly lower than for PT or PT + ICM, as for the case of open boundary conditions discussed in the main text. Notice that auxiliary Metropolis spin flips are now no longer needed to help in the equilibration of some of the configurations of some of the disorder realizations.

Table 2: First row: target value of $\Delta$. Second and third row: each entry represents a lower bound on the number of Monte Carlo sweeps necessary to decrease $\Delta$ below the value indicated in the same column for parallel tempering (PT) and parallel tempering plus isoenergetic cluster moves (PT + ICM) (data read off Fig. 2 of Ref. [56]). Fourth row: Upper bounds on the number of TNMH iterations necessary for the same purpose. The setting considered is identical to Ref. [56] (periodic boundary conditions).

| $\Delta$ | 0.25 | 0.15 | 0.05 | 0.025 |
|---|---|---|---|---|
| PT | $2^{21}$ | $2^{22}$ | $2^{23}$ | $2^{24}$ |
| PT + ICM | - | - | $2^{13}$ | $2^{14}$ |
| TNMH | 14 | 21 | 40 | 56 |

The reason for this is that by freezing a different portion of the spins at each TNMH iteration, one is now dealing with a different effective current configuration for a different effective OBC hamiltonian, at each TNMH iteration. Even though an effective current configuration may occasionally suffer from the ill-conditioning issue described in Section 3, it will not as easily stall our Markov chain thanks to the selection of a different cut at each iteration.

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
