# Peer review of "Collective Monte Carlo updates through tensor network renormalization"

_SciPost Physics, doi:SciPost Phys. 14, 123 (2023)_

## Round 3 · Referee Report · Anonymous · 2023-1-30

Strengths
The proposed tensor network Metropolis-Hastings Markov chain method is universal, versatile, has high acceptance rates and can produce collective updates.
Report
The manuscript presents a new Metropolis-Hasting Markov chain Monte Carlo method for sampling Boltzmann distributions of classical spin systems. The key idea of the approach is to use tensor network contractions to perform collective updates at each step of the evolution. The authors present benchmark computations for various cases of the two-dimensional Ising model and also make an extension to three spatial dimensions. The results indicate the algorithm is more efficient in sampling and updating than the conventional classical Monte Carlo methods. The paper is well-written, and the results are presented in a clear and concise manner. Overall, the paper is a valuable contribution to the field. I would recommend this manuscript be published in SciPost, after addressing the comments below:
1. The paper demonstrates the high efficiency of the TNMH method for sampling ferromagnetic and antiferromagnetic systems. However, data on the effectiveness of the algorithm, such as the sampling acceptance rate, is lacking for the two-dimensional triangular lattice model and the solid sphere model.
2. Most of the models mentioned in the article are described by classical Hamiltonians. Are there any effective extensions of the TNMH method for quantum Hamiltonians? such as calculating the ground state or correlation functions of a quantum system?
Author: Miguel Frías Pérez on 2023-03-08 [id 3453]
(in reply to Report 1 on 2023-01-30)
We thank the first referee for their positive assessment of the paper. Below, we address the comments that they made:
- The paper demonstrates the high efficiency of the TNMH method for sampling ferromagnetic and antiferromagnetic systems. However, data on the effectiveness of the algorithm, such as the sampling acceptance rate, is lacking for the two-dimensional triangular lattice model and the solid sphere model.
We agree with the referee that it would be interesting to explore these models further. However, we have decided to limit the scope of this paper to the conceptual and methodological aspects. This allows us to convey more clearly its message, while briefly discussing its potential applications. We demonstrate the effectiveness of the algorithm with data for many instances of the Ising model around criticality, as well as for the XY model. Section 5 is mainly meant to show that TNMH is not restricted to the models we have chosen for benchmark. As a matter of fact, we are currently investigating triangular lattices and some plaquette models.
- Most of the models mentioned in the article are described by classical Hamiltonians. Are there any effective extensions of the TNMH method for quantum Hamiltonians? Such as calculating the ground state or correlation functions of a quantum system?
We find the point raised here by the first referee very interesting. Our method can be easily adapted to study quantum systems using path integral quantum Monte Carlo. We also believe that other approaches, such as the Stochastic Series Expansion, admit a representation where our collective updates are useful, and it is a direction that we are currently exploring.
Author: Miguel Frías Pérez on 2023-03-08 [id 3454]
(in reply to Report 2 on 2023-01-31)We thank the second referee for their positive assessment of the paper. Below, we address the comments made by them.
As mentioned in the reply to the first report, we find this point very interesting. The path integral weights clearly admit a TN representation, so our algorithm should be directly applicable to models described in that formalism. We also believe that other approaches, such as the Stochastic Series Expansion, admit a representation where our collective updates are useful, and it is a direction that we are currently exploring.
As the referee points out, both quantities convey a similar message. To compare the two, we have used the data from our simulations in the two-dimensional ferromagnetic Ising model to estimate them both as a function of the temperature, at fixed system size, and viceversa: fixing a temperature around criticality and scaling the system size. The results are shown in the plots included in the response. "fig_S.pdf" (a) is analogous to Fig. 3a of the main text. In the left figure, we show the rejection rate as a function of the temperature for a fixed system size. In the right figure, we plot the ratio suggested by the referee. Both quantities deviate from their ideal value (0 for the rejection rate, and 1 for the probability ratio, marked both with black dashed lines) as we approach criticality. As expected, increasing the bond dimension at a fixed temperature, improves the accuracy of the method. Similar results are shown in the plot "fig_S.pdf" (b), the equivalent to Fig. 4a in our manuscript. There, we can see that at a fixed temperature near criticality, increasing the system size at a fixed bond dimension decreases the accuracy of the method, again by making both quantities deviate from the ideal value.
The two quantities seem to convey very similar information. However, we still believe that it is more relevant to show the rejection rate, as it is the one that directly has an implication on the ergodicity of our scheme. The ratio of probabilities is sensible to both configurations where it might be much larger than one, as well as configurations where it might be much smaller. This can lead to some confusion: the ideal prior $\tilde{\pi} = \pi$ has zero rejection rate and expectation value of the ratio one, which are the ideal values. However, there are imperfect priors that have ratios different from one. That is not the case for the rejection rate, as the only prior with rejection rate strictly zero is the ideal one.
We thank the referee for pointing this out. In the new version of the paper we have correctly introduced the notation where it is due.
We thank the referee for their comment, which improves the readability of the paper. As we have had to change the paper to the SciPost format, the figures have been rearranged. We hope that the new presentation conveys our results in a clearer way.
Attachment:
fig_S.pdf

---

## Round 3 · Referee Report · Anonymous · 2023-1-31

Strengths
* Novel general numerical method for classical statistical mechanics, combining Monte Carlo and tensor network, is presented.
* Extensive tests and comparisons against established method are provided.
* Many extensions are discussed as well.
Weaknesses
* Some additional information could be useful to the readers, as explained in the report.
* Small formatting issues.
Report
Monte Carlo and tensor networks are two cornerstones of the numerical study of many-body systems. This works presents a very interesting marriage of them for the study of classical statistical mechanics: The use of rough tensor network contraction to provide at a reasonable numerical cost approximate samples from the Gibbs distribution, which can then be used for Monte Carlo sampling with fast equilibration and decorrelation, while maintaining the overall exactness of Monte Carlo methods. The method is thoroughly tested on Ising systems on square lattices in 2D (ordered as well as disordered) and 3D, and its advantage over traditional Monte Carlo and tensor network systems is demonstrated. Concrete extensions to models with continuous variables, other lattices, and even hard spheres are given (though mostly not tested numerically). This is a very interesting development of a new algorithm which seems to balance numerical efficiency with reasonable coding effort, which is presented in a very clear way. I therefore recommend its publication, provided, however, the following comments are addressed first:
1) First, I would like to join the other Referee in requesting the Authors to comment on possible generalizations to quantum systems. An obvious way would be to rely on the mapping of a d-dimensional quantum system to d+1-dimensional classical one via Trotterization of the imaginary time axis, but there might be better ways to do so.
2) The ratio [\tilde{\pi}(\omega)/\tilde{\pi}(\omega^\prime)]/[\pi(\omega)/\pi(\omega^\prime)] between the tensor network estimate to the probability ratio between two configurations and the exact Gibbs probability ratio, and how close it is to unity, could be a useful indicator of the effectiveness of the scheme, as the Authors themselves indicate. Could the Authors plot this quantity for some of the models and parameter values they explored, and compare it with the rejection rate?
3) The notation \delta_K for the Kronecker delta seems to be used first a few lines after Eq. (15), but to be defined only after Eq. (19).
4) The manuscript contains many figures with one or two panels, and as a result in many places three figures are crowded into a single page. I believe that it would improve the formatting of the paper if such figures are merged into a smaller number of multi-panel figures.
Requested changes
See the list of comments in the report.

---

## Editorial Decision

published